# Coexistence of Railway and Road Services by Sharing Telecommunication Infrastructure Using SDN-Based Slicing: A Tutorial

**Radheshyam Singh** [1,†] , **José Soler** [1,*,†] , **Tidiane Sylla** [2], **Leo Mendiboure** [2] **and Marion Berbineau** [3]

1   Department of Electrical and Photonics Engineering, Technical University of Denmark,
    2800 Kgs Lyngby, Denmark
2   COSYS-ERENA Lab, University Gustave Eiffel, IFSTTAR, 33067 Bordeaux, France
3   COSYS Department, University Gustave Eiffel, IFSTTAR, 59650 Villeneuve d'Ascq, France
*   Correspondence: joss@fotonik.dtu.dk
†   These authors contributed equally to this work.

**Abstract:** This paper provides a detailed tutorial to develop a sandbox to emulate coexistence scenarios for road and railway services in terms of sharing telecommunication infrastructure using software-defined network (SDN) capabilities. This paper provides detailed instructions for the creation of network topology using Mininet–WiFi that can mimic real-life coexistence scenarios between railways and roads. The network elements are programmed and controlled by the ONOS SDN controller. The developed SDN application can differentiate the data traffic from railways and roads. Data traffic differentiation is carried out using a VLAN tagging mechanism. Further, it also provides comprehensive information about the different tools that are used to generate the data traffic that can emulate messaging, video streaming, and critical data transmission of railway and road domains. It also provides the steps to use SUMO to represent the selected coexistence scenarios in a graphical way.

**Keywords:** railways; SDN; ONOS controller; handover; traffic differentiation; Mininet–WiFi; OpenFlow; VLAN

## 1. Introduction

The European Railway Traffic Management System (ERTMS) is considering available modern communication technologies to redefine the communication system for railways. The new communication system is based on multiple-access wireless technologies with 5G as main its target [1] to enhance service capabilities and safety. The Future Railway Mobile Communication System (FRMCS) [2] has defined the desired specifications and user requirements [3]. A number of use-cases, for instance: voice communication to/from the controller, public emergency communication, on-train safety, critical data sharing, remote control rail engine communication, etc., related to the FRMCS system are considered in [4]. To validate the FRMCS's considered requirements and use-cases, multiple tests and functional analyses are required while developing the prototypes [5].

The empirical work presented in this paper is part of a European Union project "EU H2020 ICT 5G for FRMCS (5GRAIL)" [6]. The goal of the 5GRAIL project is to create and test FRMCS ecosystem prototypes in order to validate the initial set of FRMCS requirements and standards (FRMCS V1). In this research, we emulate the telecommunication infrastructure for different scenarios for railway and road coexistence environments that are considered in Work Package 6 (WP6) [7] based on software-defined networks. Using SDN, networks can be configured dynamically and programmatically, and this mechanism improves the performance and monitoring of the network. The technology is more like cloud computing than traditional network management due to its ability to configure networks dynamically. The network elements are controlled by a centralized SDN controller [8].

The initial steps of this research are to find out the tools and technology that can be used to develop and emulate the different scenarios for railway and road coexistence. In addition, the primary objective of this paper is to provide a detailed tutorial to develop an SDN-based application using an ONOS SDN controller that should have the potential to manage the network handover and differentiate the data traffic from railways and roads. To differentiate the data traffic, SDN-based slicing is used. Over a common infrastructure, network slicing creates multiple end-to-end virtual networks. These virtual networks are logically isolated from each other and can be customized to serve different types of services with different requirements.

These are the considered objectives of Work Package 6 of the 5GRAIL project [7]. The second objective of this paper is to provide steps and logic to develop the network topology. The network topology is created in such a way that hosts should have the mobility function, i.e., hosts can move in the given direction at an assigned speed. The purpose of this functionality is to represent the hosts as cars and trains/trams. The third objective of this empirical work is to provide steps and procedures to represent the railway and road coexistence scenario in a pictorial way using SUMO tools. Along with these issues, the following questions are considered and analyzed to provide a fruitful outcome:

1.  **Emulation:** How can emulation of trains and cars (mobility and handover scenarios) be performed in telecommunication infrastructure with the selected SDN-based tools for railway and road coexistence scenarios?
2.  **Differentiation of Data Traffic:** How are the data traffic from railways and roads differentiated, or how is the SDN application designed and developed to perform SDN-based slicing to differentiate between data packets from railways and roads?
3.  **Traffic Generation Tools:** Which tools are selected to generate different data traffic that can mimic the real-life data transmission scenario, and how can these tools be used to generate the intended data traffic?
4.  **Assessment of Developed SDN Application and Tools:** Will the considered tools and technology have the potential to emulate the coexistence scenario of railways and roads?

This paper follows the following structure: Section 2 presents some previously carried out research. Section 3 describes the selected scenarios to develop the network topology. Information about the selected tools is given in Section 4. The setup description to carry out the test is presented in Section 5. Section 6 demonstrates network topology creation using Mininet–WiFi. The selected tools for data traffic generation are presented in Section 7. Section 8 elaborates on the developed ONOS SDN slicing application to differentiate the data traffic based on VLAN tagging. Validation of the developed SDN application and demonstration of selected tools are given in Section 9. SUMO integration for train and car mobility traffic generation and visualization purposes is presented in Section 10. The conclusion of this practical work is given in Section 11.

## 2. Related Work

In this section, some previously executed research and findings are given. In [9], the authors Mininet–WiFi to compare flow-based monitoring mechanisms for the Internet of Vehicles (IoV) environment. Performance comparison of open-source SDN controllers is presented in [10] to investigate the end-to-end delay for SDN-based vehicular networks. The authors in [11] developed a network prototype to define a car as a node with Mininet–WiFi to emulate the vehicular networks. The authors in [8] demonstrated the emulation of software-defined wireless networks using Mininet–WiFi. An emulation framework is presented using Mininet–WiFi in [12] for connected autonomous vehicles. A detailed survey for connected vehicles is presented in [13]. In the paper [14], the authors investigated the handover mechanism for communication at the intersection of the coverage range of the cells for vehicular networks. They introduced multiple-access edge computing (MEC) with a roadside unit (RSU) based on SDN to improve the handover process and curtail the handover time. In paper [15], the authors used a simulation tool known as

Mininet–IoT with a Ryu–SDN controller to create a vehicular ad hoc network (VANET). They also introduced a new interface based on the global positioning system (GPS) for cars to establish communication between cars when they are not in the coverage range of access points. This proposed system has better throughput and lower delay compared to WiFi–VANET. In [16], the authors emulated the 5G-NR for the vehicular network using Mininet–WiFi-Containernet and SUMO tools.

From the research mentioned above, it can be observed that narrow research or emulation work has been carried out in the field related to vehicular networks based on software-defined networks. Based on our analysis at the time of writing this paper, we did not find any related work that considered the emulation of the coexistence scenario of railways and roads based on SDN slicing to differentiate the data traffic to/from trains and cars by sharing a telecommunication infrastructure. We can say that this is one of the first studies where the authors have considered coexistence scenarios of railways and roads using SDN-based slicing. In addition, we present a detailed tutorial about the execution of selected tools.

## 3. Selecting and Defining the Scenarios

This section provides information about the considered scenarios for railway and road coexistence.

### 3.1. Essential Parameters to Define the Scenario

The selection of scenarios to demonstrate the coexistence of railways and roads is highly complex since it depends upon multiple parameters. If we consider telecommunication infrastructure sharing, radio access network sharing parameters, and elements that are coupled with network topology components that are presented in Table 1, then the following variables are taken into the consideration to define the scenarios:

(A)  **Telecommunication Network Elements:** To define the railway and road coexistence scenario, telecommunication network parameters such as radio access network (RAN), backhaul, and core are considered as one vital parameter. Railway and road access networks can be shared or dedicated for both domains. Similarly, backhaul and core can be dedicated or shared [7].

**Table 1.** Considered combination cases for radio access and core network [7].

| | | Dedicated RAN | Shared RAN |
|---|---|---|---|
| **Radio Access Network** | **Single Technology** | R1 | R2 |
| | **Multiple Technology** | R3 | R4 |
| **Core Network** | | C1 | C2 |

Nomenclature used to represent the telecommunication infrastructure is given below:

- T1 R1C1: single-serving technology is used in the access network and each domain has its own dedicated RAN and its own dedicated core network.
- T2 R1C2: single-serving technology is used in the access network, each domain has its own dedicated RAN, and the core network is shared by both domains.
- T3 R2C1: single-serving technology is used in the access network and the RAN is shared by both domains, but both domains have their own dedicated core network.
- T4 R2C2: single-serving technology is used in the access network and each domain shares the access network and the core network.
- T5 R3C1: different-serving technology is used in the access network and each domain has its own dedicated RAN and its own dedicated core network.

- T6 R3C2: different-serving technology is used in the access network, each domain has its own dedicated RAN, and the core network is shared by both domains.
- T7 R4C1: different-serving technology is used in the access network and the RAN is shared by both domains, but both domains have their own dedicated core network.
- T8 R4C2: different-serving technology is used in the access network and each domain shares the access network and the core network.

(B) **Mobility Parameters:** The speed of the vehicles and the operating region of vehicles are also considered as shaping parameters to define the scenarios, e.g., train versus high-speed train versus urban train versus regional train versus highway versus road [7].

Nomenclature used to represent the telecommunication infrastructure is given below:

- M1: mobility of highway and tram;
- M2: mobility of highway and urban train;
- M3: mobility of highway and regional train;
- M4: mobility of highway and high-speed train;
- M5: mobility of road and tram;
- M6: mobility of road and urban train;
- M7: mobility of road and regional train;
- M8: mobility of road and high-speed train.

(C) **Topological Elements in Civil Engineering Infrastructure:** That deployed railway track components are parallel or perpendicular to roads is also an essential factor to define railway and road coexistence scenarios. This also includes transport infrastructure such as open places versus tunnels versus bridges [7].

Nomenclature used to represent the telecommunication infrastructure is given below:

- P1: railway tracks parallel to the road, open-air/bridge, the same plane;
- P2: railway tracks parallel to the road, open-air/bridge, different planes;
- P3: railway tracks parallel to the road, tunnel, same plane;
- P4: railway tracks perpendicular to the road, open-air/bridge, same plane (level crossing);
- P5: railway tracks perpendicular to the road, open-air/bridge, different planes;
- P6: railway tracks perpendicular to the road, tunnel, different planes.

(D) **Services:** Services and applications defined for railways and roads play an important role in defining the scenario. For instance, speed-limit-monitoring applications, traffic rule violation monitoring applications for vehicles, applications to monitor the railway tracks to locate cracks, etc.

### 3.2. Selected Scenario

Considering all the parameters and dependent variables mentioned above, approximately 400 scenarios can be defined related to railway and road coexistence scenarios [7]. For this practical work, five different feasible scenarios are considered that are designed, developed, and executed using the considered tools. The selected scenarios have nomenclature SX(Y)Z, where: S represents the term "Scenario", X represents the "Telecommunication Network Parameters", Y represents the "Mobility Parameters", and Z represents the "Topological Elements in Civil Engineering Infrastructure".

1.  **S1(5/6)1: Different Access Network and Different Core, Single Serving Technology, Track Parallel to Road:** This scenario is considered the baseline scenario to investigate the coexistence of railway and road scenario telecommunication services infrastructure. In this scenario, both domains, i.e., railways and roads, have their own dedicated radio access network (RAN) and dedicated core. Considered access points

and cores work on a single technology/radio frequency. Along with this, railway tracks are kept parallel to roads [7].

2.  **S1(5/6)4: Different Access Network and Different Core, Single Serving Technology, Track Perpendicular to Road:** In this considered scenario, the network parameters are similar to scenario S1(5/6)1, i.e., railways and roads have dedicated radio access networks with dedicated cores, but in this scenario, railway tracks are perpendicular to roads [7].

3.  **S2(5/6)1: Different Access Network and Shared Core, Single Serving Technology, Track Parallel to Road:** In this scenario, railway and road domains have different radio access networks, and both domains share backhaul and core network infrastructure. In this considered scenario, railway tracks are perpendicular to roads [7].

4.  **S4(5/6)1: Shared Access Network and Shared Core, Single Serving Technology, Track Parallel to Road:** In this scenario, railway and road domains share the radio access network along with backhaul and core network infrastructure. Railway tracks are parallel to roads [7].

5.  **S4(5/6)4: Shared Access Network and Shared Core, Single Serving Technology, Track Perpendicular to Road:** In this considered scenario, the network deployment infrastructures are similar to those of scenario S4(5/6)1, but in this case, railway tracks are kept perpendicular to roads [7].

## 4. Selected Tools

Based on the objectives presented in Section 1 for this research, the following are the key requirements that should be fulfilled by the selected tools:

1.  The selected tools should have the capability to define the end nodes in such a way that the user can configure the mobility of the host in a selected direction with an assigned moving speed. The selected tool should also be able to define the quantity and frequency of the end points.

2.  Using the selected tools, users should have the possibility to define multiple wireless network interfaces for end nodes.

3.  Using the selected tools, there should be the possibility to generate different kinds of data traffic from the end nodes, compliant with different types of services: for example, messaging, critical data communication using messaging services or video streaming, and voice communication for operational purposes.

4.  Radio channel characteristics can be defined to mimic real characteristics of radio links such as packet loss, network jitter, and delays.

5.  The defined end nodes should have the capability to connect to an available WiFi-based network.

6.  The selected tools should have the ability to define different network topologies for wireless access points as well as for fixed network entities and provide SDN-based OpenFlow interfaces. In addition, from a software-defined network perspective, a defined network component's behavior can be controlled and managed by a network controller.

7.  To differentiate the data traffic from different network end nodes, selected tools should have the ability to support virtual local area network (VLAN)-based tagging/untagging, and they should support tag-based forwarding.

8.  The selected tools have the ability to emulate cellular-based network connectivity, principally 5G.

9.  It would be nice to have a tool that can graphically represent the network emulation for the different selected scenarios.

Based on the requirements considered above, the Open Network Operating System (ONOS) SDN controller [17] is selected to programmatically control the network topology, and Mininet–WiFi [18] is selected to define the network topology. The tool SUMO [19] is considered for the graphical representation of the selected scenario. Figure 1 shows the

selected tools with their key properties that are considered to emulate the railway and road coexistence scenarios.

- ONOS has capability to control the behavior of network elements programmatically .
- It supports VLAN tagging/un-tagging and tag-based routing.
- It also supports movement tracking and handover of endpoints.

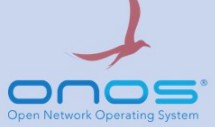

- Has possibility to define different network topologies for wireless access points and for fixed network.
- It can define speed, position and quantity etc. of endpoints.
- Has possibility to generate data traffic from endpoints.
- It can define the radio channel characteristics.
- Wi-Fi-based connectivity
- It has ability to provide Software Define Network (SDN) (OpenFlow) interfaces.

- SUMO tool has capability to integrate with Mininet-WiFi to represent the hosts as car, train, tram or bus on the selected geo-map.

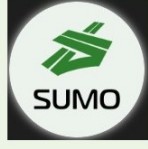

**Figure 1.** Selected tools.

### 4.1. ONOS SDN Controller

The ONOS software-defined network controller is an open-sourced SDN and network function virtualization (NFV) controller. A simplified programmatic interface makes ONOS an ideal platform for operators searching to build innovative and advanced network services. ONOS has the ability to configure and control the network by programming the functionality and reducing network protocol implementation requirements. The ONOS cloud controller integrates intelligence, enabling end-users to easily create new network applications without having to change the data plane [17].

### 4.2. Mininet–WiFi

Mininet–WiFi [18] is a software-defined network emulator. It is a branch of Mininet [20] embedded with additional functionalities such as the ability to define and configure WiFi access points and nodes with moving capability based on Linux wireless driver and simulation driver 80211_hwsim [21]. Using Mininet–WiFi, users can define different network topologies, where host/nodes can be defined with multiple wireless interfaces. Along with these, Mininet–WiFi supports defining radio parameters such as operating frequency channel, propagation model, coverage range, and transmission power (Tx). The network topologies developed with Mininet–WiFi have the potential to be controlled programmatically based on OpenFlow protocol versions 1 through 5. Since it works on the Linux wireless driver 80211_hwsim, it does not have the ability to emulate 5G-based connectivity. Therefore, it can be observed that Mininet–WiFi fulfills the requirements (1-7) that are taken into consideration for this practical work. Only Requirement 8 is not covered by this selected tool. We discuss this in the Conclusions (Section 11). Mininet–WiFi installation files and processes are available at [18,22].

### 4.3. SUMO

Simulation of urban mobility, commonly known as SUMO [19,23], is an open-sourced traffic simulator used to design and visualize the mobility of vehicular networks. SUMO supports features such as multimodal and continuous mobility of selected nodes/stations. Using SUMO, users can define the speed and quantity of selected nodes (cars, train, tram, bicycle, etc.). Based on the user's interest, the simulation area can be extracted directly from the open street map, where users can select the intended simulation area and download the simulation map files. Further, an additional feature can be added that shows the map area with assigned access points and nodes in a graphical manner. The authors of [24,25] used SUMO for visualization, modeling, and defining nodes in traffic routes. Therefore, SUMO is considered to fulfill Requirement 9 mentioned above.

## 5. Setup Description

Figure 2 represents the test setup overview. All the selected tools that are considered to emulate the railway and road coexistence scenarios are installed on a virtual machine. The considered network topology is created using Mininet–WiFi for road and railway coexistence scenarios. To control the functionality of the network topology, an SDN application is developed, installed, and activated for the ONOS SDN controller. The SDN application is developed to support the moving of end nodes and the inter-cell handover of created nodes in a defined virtual space. It also has the ability to differentiate the data traffic based on VLAN tagging. Detailed information about this developed SDN application is elaborated in Section 8. SUMO is integrated with Mininet–WiFi to graphically represents the movement of network nodes on an open street map.

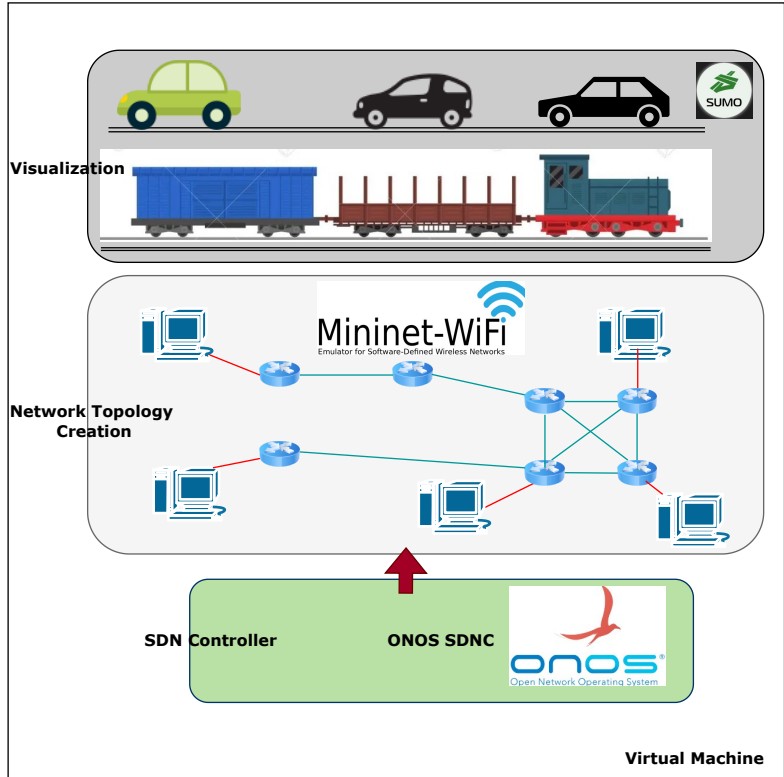

**Figure 2.** Test setup overview.

## 6. Network Topology Creation Using Mininet–WiFi

Before starting the validation of the selected tools and ONOS applications, the network topologies are created based on the scenarios explained in Section 3.2 for the coexistence of railway and road environments.

In this research, we implement and investigate all five scenarios presented in Section 3.2. For demonstration purposes to explain the tutorial to emulate the coexistence scenario of railways and roads, S2(5/6)1 and S4(5/6)4 scenarios are selected. The objective of this section is to provide all the necessary instructions to create the selected network topologies.

### 6.1. Network Topology S2(5/6)1—Different Access Network and Shared Core, Railway Track Parallel to Road

A Python script is written to create a network topology with Mininet–WiFi where trains and cars have different access networks, both domains shared the core network, and railways have parallel tracks to roads. Figure 3 shows the S2(5/6)1 scenario, where an ONOS SDN controller programmatically controls the forwarding elements of the topology. The network switches and access points are SDN-based devices and they are operated and

controlled via OpenFlow protocol. Host Car1 represents a car, and host Train1 represents a rail. Access points ap1 and ap2 are defined for roads, and ap3 and ap4 are defined for railways. Access points ap1 and ap3 are connected to network switch S11, and ap2 and ap4 are connected to switch S33. The switch S22 is connected to S11 and S33. The host "RailServer" is defined as a railways service server, and "CarServer" is defined as a road service server; both servers are connected to switch S22. In this simple network topology, switch S22 is the core network switch, and S11 and S33 are the edge switches.

The network topology can be complex with more core and edge network elements. For the demonstration, a simple topology is selected.

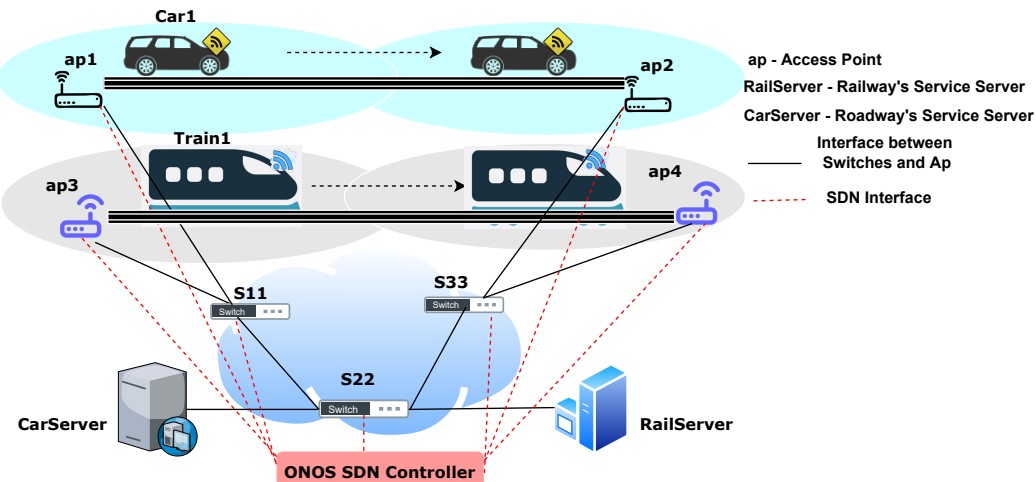

**Figure 3.** S2(5/6)1: Different Access Network and Shared Core, Track Parallel to Road.

In Figure 3, we can see that railways and roads have different access networks but a shared core network. To design this scenario, a topology development code is written in Python. The source code of this complete project is available at [26].

As we mentioned, for this practical work, the ONOS SDN controller is used. To define the ONOS as a remote controller as the element of the topology, the code given in Listing 1 is used as the Python topology code. To use the remote controller in Mininet–WiFi, port 6653 is assigned to the ONOS controller defined as "c1". Since Mininet–WiFi and the SDN controller are executed from the same virtual machine/laptop, IP '127.0.0.1' is assigned to the controller.

**Listing 1.** Adding Remote Controller ONOS.

```
''Create_a_network.''
net = Mininet_wifi( topo=None,
               build=False,
               ipBase='192.168.0.0/24')
info( '***Adding_Remote_controller\n' )
c1=net.addController(name='c1',
   controller=RemoteController,
            ip='127.0.0.1',protocol='tcp',port=6653)
```

After adding the ONOS controller, network switches and access points are defined. To define the switches, *net.addSwitch()* with "OpenFlow13" protocol is used, as shown in Listing 2. Using *net.addSwitch()*, three Open Virtual Kernel switches (OVS), S11, S22, and S33, are defined that work on "OpenFlow13" protocol. The command *net.addStation()* is used to define the cars and trains because it has the capability of defining the moving functionality and speed of the host. The hosts "CarServer" and "RailServer" are defined using the *net.addHost()*. The *args()* function is used to define the initial location of the moving nodes.

**Listing 2.** Defining the Moving Host and Server Host.

```
info ( '*** Add_switches \n')
s11 = net.addSwitch('s11',cls=OVSKernelSwitch,protocols="OpenFlow13")

info(''*** Creating nodes\n'')
Train1_args , Car1_args = dict(), dict()
if '-s' in args:
    Train1_args['position'] , Car1_args['position']
    = '30,10,0' , '30,320,0'

Train1 = net.addStation('Train1', mac='00:00:00:00:00:01',
        ip='192.168.7.101/24', position='30,10,0', **Train1_args)

Car1 = net.addStation('Car1', mac='00:00:00:00:00:02',
        ip='192.168.0.201/24', position='30,320,0',**Car1_args)

info ( '*** Add_hosts_for_Service_Server\n')
CarServer = net.addHost('CarServer', cls=Host,
            ip='192.168.0.204/24', mac='00:00:00:00:00:08')
RailServer = net.addHost('RailServer', cls=Host,
            ip='192.168.7.104/24', mac='00:00:00:00:00:07')
```

To define the access points in Mininet–WiFi, *net.addAccessPoint()* is used as shown in Listing 3. It provides the ability to define and assign parameters such as the position of the access point, mode, operating channel, transmitting power in dBm, and coverage range in meters. Using a similar mechanism, four access points, ap1, ap2, ap3, and ap4, are defined for the selected scenario. The access points are assigned with different operating frequency ranges. The parameters *mode* and *channel* define the operating frequency range to broadcast and receive signals. The available modes for WiFi are b/g/n on a 2.4 GHz network and a/ac/n on a 5 GHz network [27]. The *net.setPropagationModel()* function is used with path loss exponent 5.

**Listing 3.** Defining the Access Point.

```
ap1 = net.addAccessPoint('ap1', ssid='ssid-ap1', mode='a',
        channel='36', position='100,300,0',
        'protocols': 'OpenFlow13',
        'txpower':'49dBm','range': 110 )

net.setPropagationModel(model="logDistance", exp=5)

info(''*** Configuring_wifi_nodes\n'')
net.configureWifiNodes()
```

After creating the network switches, hosts, and service server, the connectivity links are created using *net.addLink()* as shown in Listing 4. The parameter *bw* is used to define the bandwidth in Mbps of the connecting link. The line of codes present in Listing 4 create links between the hosts, switches, access points, and nodes. Access points ap1 and ap3 are connected to switch S11, and ap2 and ap4 are connected to switch S33.

**Listing 4.** Defining the connectivity.

```
info('''***Creating_links\n''')

s11s22 = {'bw':1000}
net.addLink (s11, s22, cls=TCLink , **s11s22)
s22s33 = {'bw':1000}
net.addLink (s22, s33, cls=TCLink , **s22s33)

CarServers22 ={'bw':500}
net.addLink (CarServer, s22, cls=TCLink , **CarServers22)
RailServers22 = {'bw':500}
net.addLink (RailServer, s22, cls=TCLink , **RailServers22)

s11ap1 = {'bw':500}
net.addLink (s11, ap1, cls=TCLink , **s11ap1)
s11ap3 = {'bw':500}
net.addLink (s11, ap3, cls=TCLink , **s11ap3)
s33ap2 = {'bw':500}
net.addLink (s33, ap2, cls=TCLink , **s33ap2)
s33ap4 = {'bw':500}
net.addLink (s33, ap4, cls=TCLink , **s33ap4)
```

Mininet–WiFi has a method *net.plotGraph()* to show the position of defined hosts and access points on a graph. Using this method, the user can define the range of x- and y-coordinates for the plot limits, as shown in Listing 5. Figure 4 shows the graphical representation of the selected topology; we can clearly see access points ap1, ap2, ap3, and ap4 and defined nodes. The developed network topology with assigned switches, access points, connected hosts, and nodes/stations can be seen on the ONOS graphical representation platform as shown in Figure 5. The network topology is built using *net.build()*, and access points are started using *start()*. Mininet–WiFi also provides a command line interface (CLI) using the function *CLI(net)*, which enables interaction with any element of the developed topology. To stop the network, the *net.stop()* function is used. In the code below, "c1" is the name assigned to the ONOS SDN controller.

**Listing 5.** Defining the Plot Limits of Graph and Start the Access Points and Switches.

```
if '-p' not in args:{
    net.plotGraph (max_x=450, min_x=-50, min_y=-50, max_y=450)}

info('''***Starting_the_network\n''')
net.build()
info( '***Starting_the_controllers\n')
for controller in net.controllers:
    controller.start()
ap1.start([c1])
ap2.start([c1])
ap3.start([c1])
ap4.start([c1])
net.get('s11').start([c1])
net.get('s22').start([c1])
net.get('s33').start([c1])

info('''***Starting_CLI\n''')
CLI(net)
```

```
info('''***Stopping_network\n''')
    net.stop()
if __name__ == '__main__':
    setLogLevel('info')
    topology(sys.argv)
```

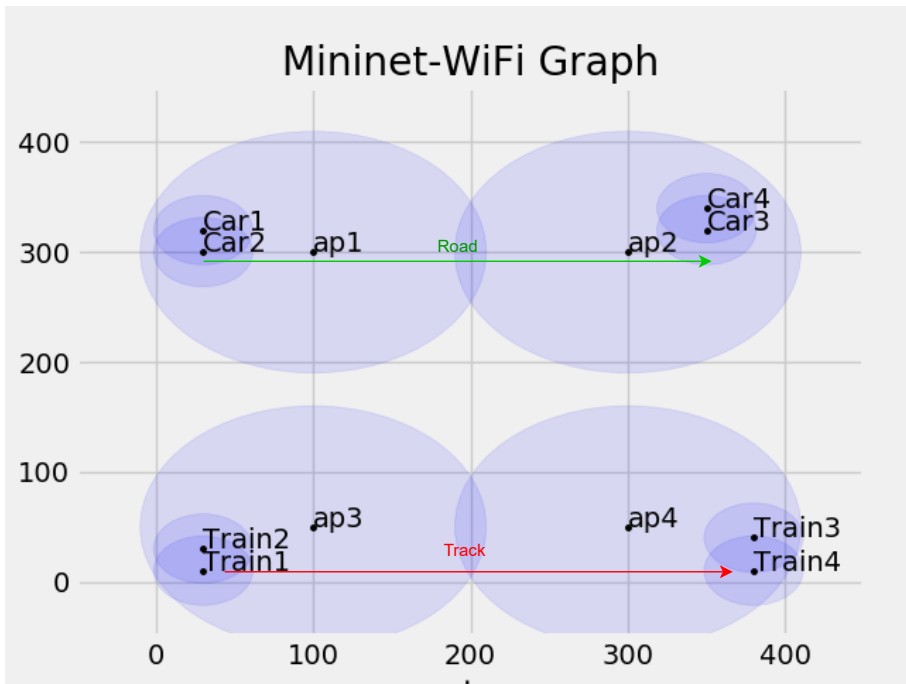

**Figure 4.** Hosts and access points: Mininet–WiFi graph.

Using the above-mentioned methods, functions, and procedures, users can emulate any salable and complex topology. Figure 5 represents the topology created using the Python script for the S2(5/6)1 scenario represented by the ONOS SDN controller's graphical user interface (GUI), where hosts Car1 and Train1 are defined with moving capability.

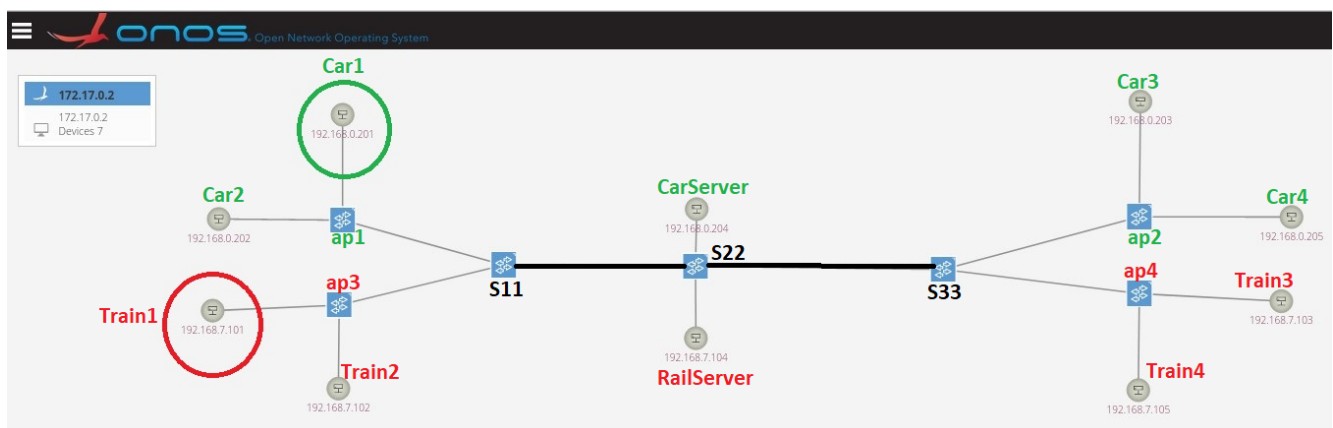

**Figure 5.** S2(5/6)1: Different Access Network and Shared Core, Track Parallel to Road Topology: ONOS screenshot.

*6.2. S4(5/6)4: Shared Access Network and Shared Core, Track Perpendicular to Road*

To design the S4(5/6)4 scenario, a similar Python script is written using the methods and functions mentioned in Section 6.1, and a network topology is developed in such a way that access points and the core network are shared by both railway and road domains.

In this selected scenario, railway tracks are perpendicular to roads. To emulate the railway tracks being perpendicular to the road, the y-coordinate of the moving node Car1 is kept constant (y = 70) while moving in the defined virtual space, and the x-coordinate of node Train1 is kept constant (x = 130) while moving in the defined virtual space, as shown in Figure 6.

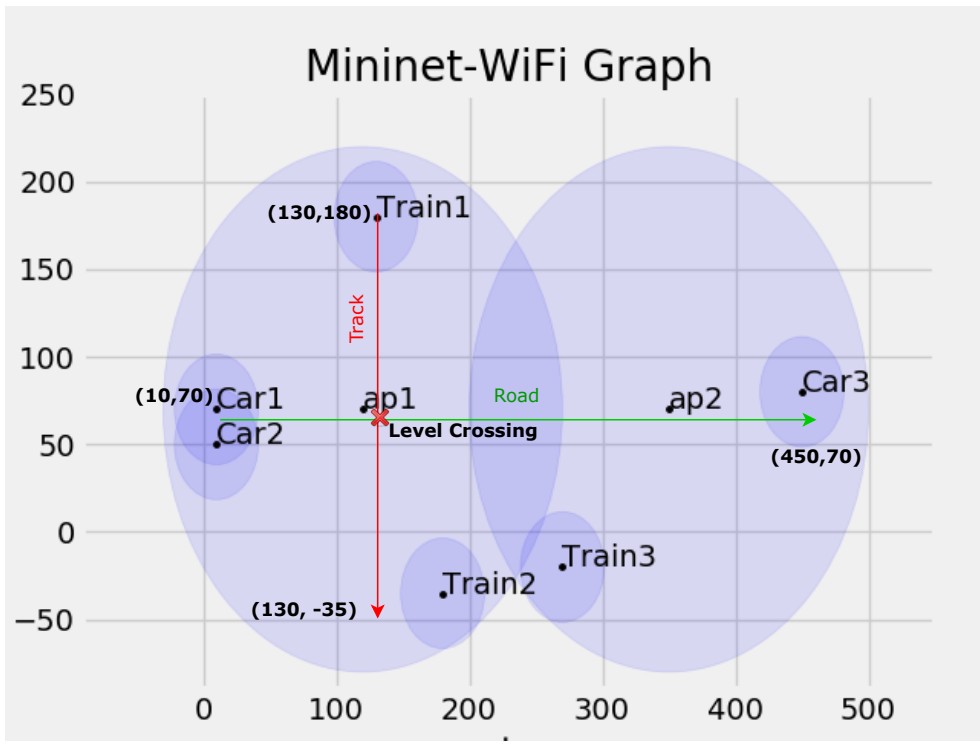

**Figure 6.** S4(5/6)4 hosts and access points: Mininet–WiFi graph.

A pictorial representation of this scenario is shown in Figure 7, where we can see that access points ap1 and ap2 are shared by both the domains, and the railway track is perpendicular to the road. This particular scenario for railway and road coexistence represents the "Level Crossing" at cross points.

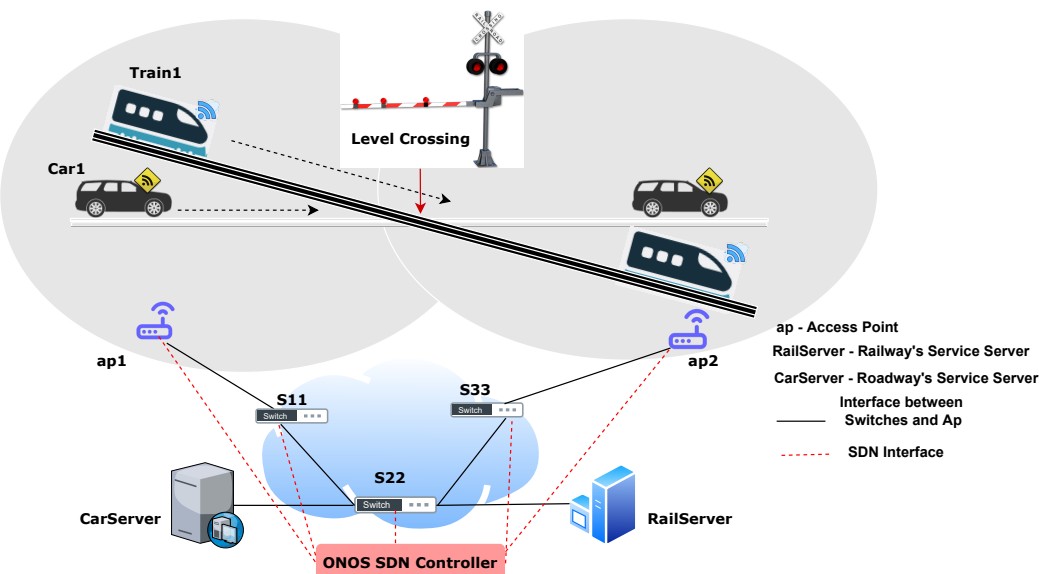

**Figure 7.** S4(5/6)4: Shared Access Network and Shared Core, Track Perpendicular to Road.

Figure 8 shows the network topology for scenario S4(5/6)4 created with Mininet–WiFi. Car1, Car2, Train1, and Train2 are connected to access point ap1, whereas Car3 and Train3 are connected to access point ap2. Hosts CarServer and RailServer are connected to switch S22. Similar to topology S2(5/6)1, in this topology, Car1 and Train1 are defined with moving functionality. Comparing Figures 5 and 8, we can see that in Figure 5 both domains have dedicated access networks, but in Figure 8, both domains, i.e., railways and roads, share the same access network.

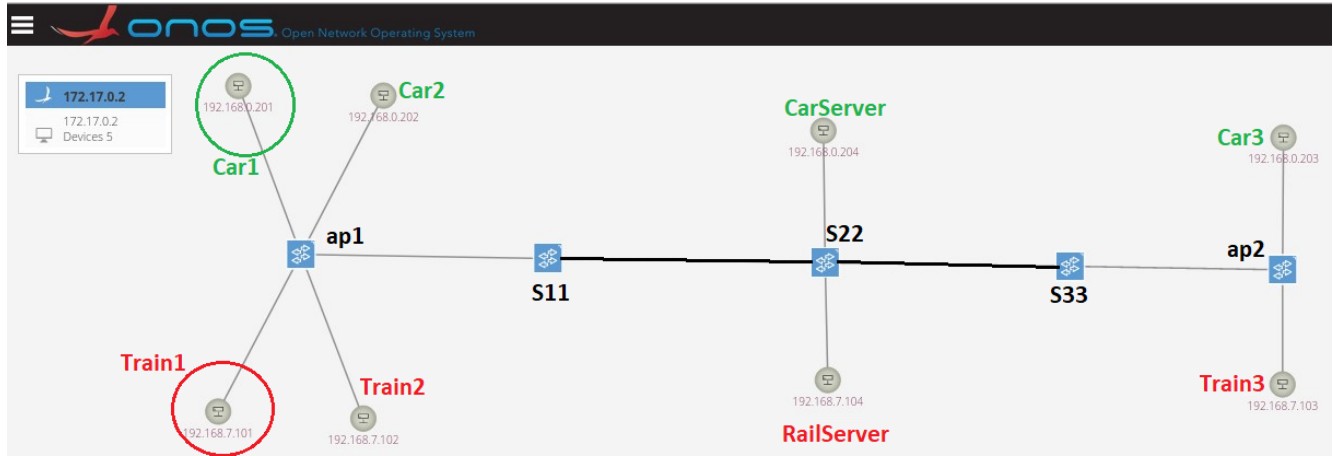

**Figure 8.** S4(5/6)4: Shared Access Network and Shared Core, Track Perpendicular to Road Topology: ONOS screenshot.

## 7. Selected Tools to Generate Data Traffic to Validate the Scenarios

To map the data traffic generation that can define the scenarios mentioned in Table 2, different tools are analyzed and selected. These mentioned scenarios are taken from the documentation of 5GRail project Deliverable D6.1 [7].

This section provides information about the selected tools that are used to generate the data traffic. Along with this, it also explains the procedure to install these tools and which parameters should be selected to generate the desired data streams.

**Table 2.** Selected tools to generate different kinds of data traffic for compliance with real-case scenario [7].

| Scenario | Considered Tool to Demonstrate the Scenario | Tool Information |
| --- | --- | --- |
| Voice Communication for Operational Purposes<br>Standard Data Communication | iperf3 | Iperf3 can send UDP and TCP packets from one host to another. |
| Critical Data Communication<br><br>Very Critical Data Communication<br><br>Messaging | Scapy | Using Scapy, we can define our data packets and send them to the network. Using Scapy, messaging and critical data communication is demonstrated. |
| Critical Video Communication for Observation Purpose<br><br>Very Critical Video Communication Associated with Train Safety | VLC Player | To demonstrate video transmission from train or car to the assigned server, VLC player is used. |
| Measure Network Quality of Services (QoS) | MTR | MTR tool has the capability to measure the latency, packet loss, and jitter of the network. |

The following tools are used and tested to generate different kinds of data traffic compliant with real-case scenarios:

1. **iperf3:** Iperf3 is a network performance measurement tool. It is able to execute on Linux, Unix, Android, macOS, and Windows platforms. It works in client and server mode functionality. It has the ability to generate user datagram protocol (UDP) and transmission control protocol (TCP) packets from one host/client and send them to another host/server. It generates a packet data stream to measure the throughput, bandwidth, and packet loss between two hosts [28,29].

   The installation file and documentation to use ipef3 are given in the link [29]. The iperf3 tool is used to send and receive UDP and TCP data packet streams to demonstrate voice communication and standard data communication for operational purposes between car-to-car and assigned car service servers or train-to-train and assigned rail service servers. Table 3 shows the commands used to generate the UDP/TCP data traffic from the client host to the server host.

2. **Scapy:** Scapy is an interactive packet manipulation tool with a CLI, and its API is implemented using Python. Therefore, this tool can be used through CLI or by importing it to Python scripts. Using this tool, users can create, forge, or decode packets from a wide range of protocols and send them to the network. It can also capture data packets, match criteria, and reply to requests [30]. Along with these functions, most classical tasks can be handled easily, including scanning, probing, and detecting networks. Scapy has the ability to run on Linux, macOS, and Windows systems. The principal advantage of Scapy is that it offers a technique to modify and create network packages at a low level by leveraging available network protocols and configuring them based on the user's needs [31]. Therefore, Scapy is used in this practical work to create a data packet to indicate critical data communication. In addition, it is also used to create special messages associated with emergencies or to convey any kind of information to the rail service server or car service server.

   Scapy is installed in a Linux environment using the specific instances shown in the Table 4. A demonstration of using Scapy to generate data packets associated with critical data communication or messages is presented in Section 9.7 of this paper.

**Table 3.** Iperf3 commands to generate UDP/TCP data packets [29].

| Command | Explanation |
|---|---|
| 1. iperf3 [-s ǀ -c] [ options ] | |
| 2. iperf3 –s –p <port number> | Where **-s** represents host as server, **-c** represents host as client, **–u** represents UDP packets, **-b** is used to set the bitrate, **-i** option allows setting the reporting interval time in seconds; e.g., iperf3 -c 10.0.0.7 -i 2 |
| 3. iperf3 –c <server IP/host IP> -u –p <port number> -b <bitrate> | |

**Table 4.** Scapy installation commands [32].

| Options | Linux Command |
|---|---|
| 1. Install default Scapy | pip install scapy |
| 2. Install Scapy with Python | pip install - -pre scapy[basic] |
| 3. Install Scapy with dependencies | pip install  - -pre scapy[complete] |

3. **VLC Player:** VideoLAN Client, generally known as VLC, is open-source, compact media player software developed by the VideoLAN project. VLC is capable of streaming and receiving media files (video/audio). In addition to DVD video and video CD formats, VLC supports a wide variety of audio and video compression methods and media streaming protocols. It can be installed and used on any desktop system as well as on smartphones [33,34].

   To demonstrate video or critical video communication in railway and road coexistence scenarios, VLC Player is used to transmit video from a train to the RailServer and

from a car to the CarServer. A detailed process for streaming a video from one host to another or to a local system is presented in Section 9.8 of this paper.

4. **MTR:** Matt's Traceroute (MTR) is a network performance monitoring tool. Ping is one of the most widely used network diagnostics tools to figure out network reachability from one host/network entity to another. Using ping, a user sends ICMP packets from one system/host to another, and after getting the ICMP packets, the destination host/system sends an echo reply. This echo reply informs about the availability of the destination host, end-to-end delay, and packet loss, whereas the traceroute provides information about the path between the sender and receiver host. MTR utilizes both ping and traceroute functionality [35,36].

The significant reason to use the MTR tool is that it allows measurement of the data packet loss and jitter of the network. Therefore, we can say that MTR measures the quality of the network path. For installing the MTR in the Ubuntu environment, use the following command: *sudo apt-get -y install mtr*. Table 5 shows the commands and options to measure the loss and jitter of the network. There are more options available to measure packet data losses and jitter with the mentioned commands, which can be found in [35].

**Table 5.** MTR data traffic generation to measure the loss and jitter of networks [35].

| Command | Available Options | Use |
|---|---|---|
| 1. mtr –r –n –c <number of data packets> -T –P <port number> <server IP/Host IP> | Where **r** reports print; **c** defines the number of Packets; **n** disables the DNS or no DNS option; **T** shows TCP data; **u** shows UDP data; | Calculate the loss in percentage |
| 2. mtr –r –n –o ''L BAWV MI'' <server IP/Host IP> | Where **L** shows loss ratio; **B** shows min/best RTT (ms); **A** shows avg RTT (ms); **W** shows Mmx/worst RTT (ms); **V** shows standard deviation; **M** shows jitter mean/avg; **I** shows interarrival jitter | Calculate the jitter in ms |

## 8. ONOS SDN Application for Data Traffic Slicing

The most significant task of this empirical work is to design and develop SDN applications capable of fulfilling the following objective:

- Supports handover/moving capability of nodes/hosts;
- Differentiates the data traffic based on VLAN tagging/slicing;
- Is scalable to support any kind of network topology.

Keeping the above-mentioned objectives in consideration, an SDN data-forwarding application is developed using ONOS JAVA APIs [37] and deployed in the ONOS controller. The developed ONOS forwarding application has the potential to enable network slicing and to differentiate the data traffic to/from railways and roads. This implies that trains can only communicate with trains and assigned rail service servers; similarly, cars can only communicate with other cars and assigned car service servers. Along with this, additional functionality is added to manage the moving and handover of nodes between the assigned access points/cells. The application decides whether a data packet should be forwarded or dropped between the nodes, switches, and access points. Figure 9 represents the different steps of the developed ONOS application. After installing and activating the application, it initializes the "packet processor". The packet processor is an ONOS API that allows defining of the header context of packets and activates the developed applications. Two arrays are created in the application that contain the IP addresses of hosts and nodes. The first array contains the IP addresses of all cars, and the second array contains the IP addresses of all rails.

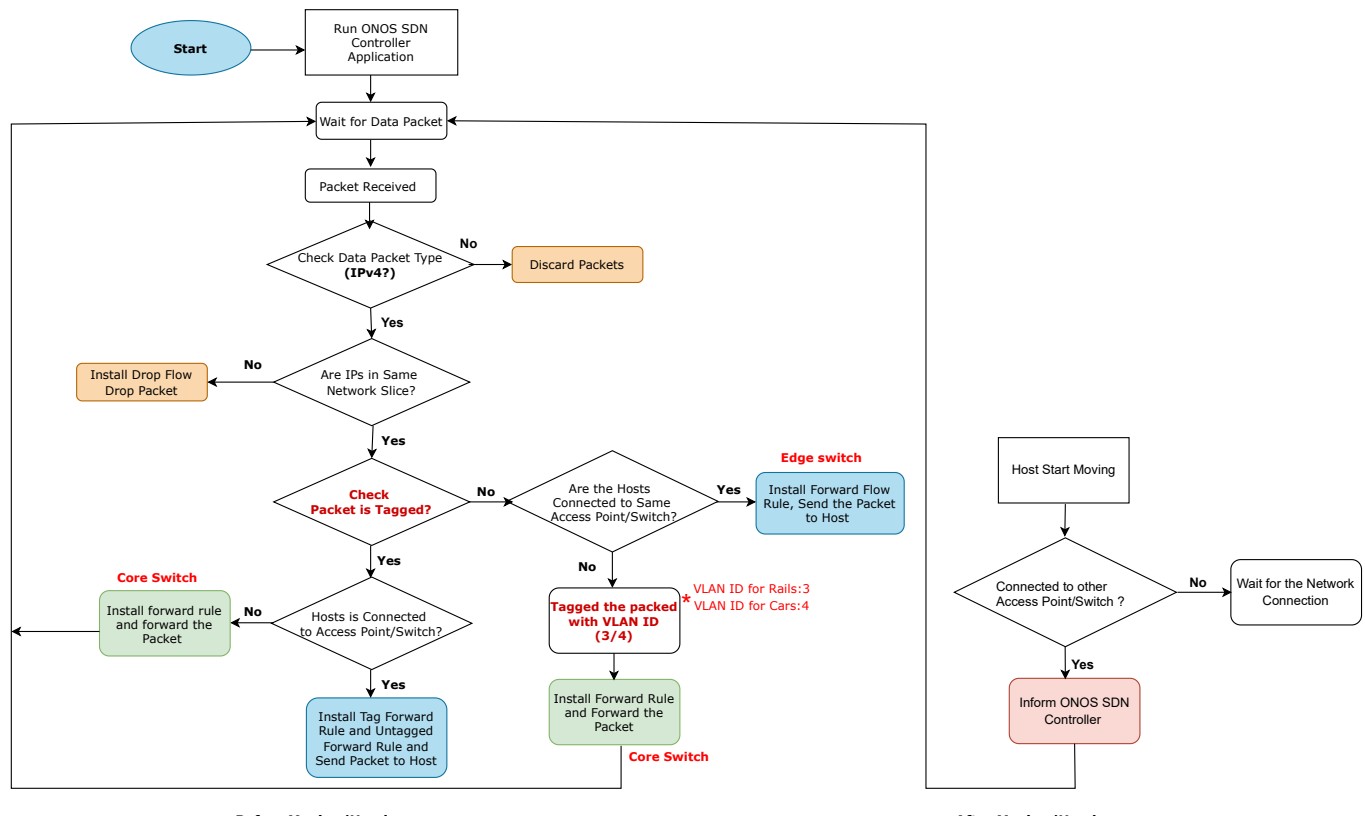

**Figure 9.** Flow diagram of SDN application.

The developed ONOS application is designed to support IPv4 data packets. When an IPv4 packet arrives at any access point/switch, it looks into its forwarding table/rules, and if there are no forwarding rules for that source and destination IP pair at the current switch/access point, the packet is sent to the ONOS controller for processing as an Open-Flow "PacketIn" message. The application checks whether both source and destination IPs are in the same network slice (trains or cars), and if they are not, the packet drop rule is installed using the OpenFlow13 protocol at the current switch/access point. This implies that any traffic between cars to trains and to their assigned service server is disabled.

If both source and destination IP pairs are in the same network slice, the application checks whether the data packet is tagged with a VLAN ID or not. If the data packet is not tagged with VLAN, and source and destination hosts are connected to the same access point/switch, the controller installs the forwarding rule using OpenFlow13 protocol at the current access point/switch, and the data packet is sent to the destination host/node. If the data packet is not tagged with a VLAN ID, and the source and destination host/node are connected to the same access point/switch, the data packet is tagged with a VLAN ID based on the network slice. For this forwarding application, the number 3 is used as the VLAN ID to tag the data packets from/to railways/rails, and the number 4 is used as the VLAN ID to tag the data packets from/to roads/cars. After tagging the data packet, the forwarding tag rule and forwarding untag rule are installed using the OpenFlow13 protocol at the current access point/switch for the IP pairs, and the data packet is forwarded to the next switch. If the data packet is tagged with a VLAN ID and the source and destination host/node are connected to the same access point/switch, the forward tag rule and untag rule are installed for the given source destination IP pair at the current access point/switch. If the data packet is tagged with a VLAN ID and the host/node is not connected to the same access point/switch, the forward rule is installed at the current access point/switch and the data packet is forwarded in the network.

If nodes move from one location to another and connect to the nearest assigned access points, the developed application informs the controller via a "PacketIn" message about the position of the nodes and connected access points.

*VLAN Tagging*

VLAN tagging is a mechanism that allows the creation of multiple networks at Layer 2 of the core network. VLAN tagging is carried out by assigning a VLAN ID to the data header as an additional element to the Ethernet header of a packet [38]. The assigned tag can be used as a filtering decision factor for the forwarding operation at switches/access points. The VLAN tag defines which side of the network part a data packet belongs to by matching the tag of the data packet header.

Figure 10 shows data packet tagging with the VLAN mechanism used in the developed ONOS application. Switches S1, S2, S7, and S8 are the edge switches, and S3, S4, S5, and S6 are the core switches. Access points ap1, ap2 and ap3 are the access network elements for cars, and ap4, ap5, and ap6 are the access network elements for trains. When a data packet is sent from the car to CarServer, access point ap1 tags the data packet with VLAN ID: 4 (cars' slice) and forwards the packet to the next switch S3. Since the data packet is already tagged with a VLAN ID, S3 matches the VLAN ID and forwards the data packet to S5; it also matches the VLAN ID and sends the data packet into the network. When this data packet arrives at the edge switch S7 where the destination host CarServer is connected, it removes the VLAN ID and sends the data packet to the CarServer. A similar working principle is followed for the train's data. The access point ap4 tags the data packet with VLAN ID: 3 (trains' slice). Core network switches S4 and S6 match the VLAN ID of the data packet and send the packet into the network. When this data packet arrives at switch S8, it removes the VLAN ID and sends the data packet to the host rail server.

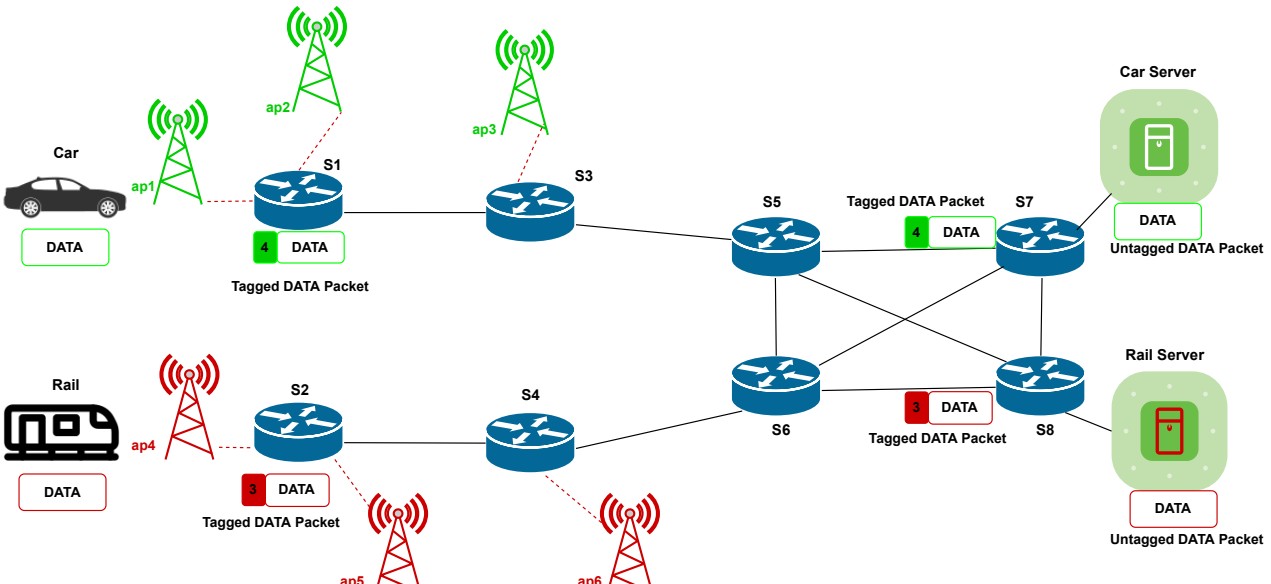

**Figure 10.** Data packet tagging.

## 9. Validation of Developed SDN Application and Selected Tools

This section provides the procedure and logic to validate the selected tools that are used to generate the data traffic. Along with this, it also investigates the developed ONOS application and analyses its functionalities such as handover and traffic differentiation capabilities based on VLAN tagging.

*9.1. Handover/Mobility (Speed/Direction/Providing Coordinates)*

To test the handover scenario, the first criteria is to design a network topology where hosts should have the capability to connect to wireless networks (WiFi) and users can define the mobility and speed of the moving host in a certain direction. These criteria can be fulfilled by Mininet–WiFi since it supports the mobility model and Mininet–WiFi Python API integrates a new method *net.addStation()* that authorizes users to define nodes that can move around in the defined virtual space, as mentioned in Section 6.1. By developing a predefined mobility model, Mininet–WiFi can also move the nodes automatically in the defined virtual space as soon as the emulation scenario starts running. Mininet–WiFi supports the following mobility models: RandomDirection, RandomWalk, RandomWayPoint, GaussMarkov, RefrencePoint, TruncatedLevyWalk, and TimeVariantCommunity [18,22].

To demonstrate the handover and moving for railway and road coexistence environments, scenario S2(5/6)1 is selected, which is described in Section 6.1 of this paper.

In the above-given Listing 6, two nodes, Car1 and Train1, are defined with moving capability. The *speed* parameter can be any feasible number greater than zero. For demonstration purposes, we keep the speed at 3 m/s. In this scenario, to define the moving and handover of nodes, the lines of codes given in List 8 are used. Node Car1 is connected to ap1 and starts moving towards ap2 after 60 s. Similarly, Train1 is initially connected to ap3 and starts moving towards ap4. The mobility of nodes starts using the *net.startMobility()* function and stops by using the *net.stopMobility()* function as shown in Listing 7. The *start* parameter initializes the starting time of the node's mobility, and*stop* initializes the stop time of the node's mobility. While moving, the ping command is executed using CLI from Car1 to CarServer and Train1 to RailServer, as shown in Figure 11.

**Listing 6.** Defining the Host With Their Moving Speeds.

```
Train1  =net.addStation('Train1',
        mac='00:00:00:00:00:01', ip='192.168.7.101/24', speed=3)
Car1    =net.addStation('Car1',
        mac='00:00:00:00:00:02', ip='192.168.0.201/24', speed=3)
```

**Listing 7.** Defining Moving Functionality of Hosts.

```
if '-s' not in args:
        net.startMobility(time=0)
        net.mobility(Train1, 'start', time=60, position='30,10,0')
        net.mobility(Car1, 'start', time=60, position='30,320,0')
        net.mobility(Train1, 'stop', time=70, position='380,60,0')
        net.mobility(Car1, 'stop', time=70, position='350,250,0')
        net.stopMobility(time=75)
```

Both of the selected nodes are pinging their assigned service server so that we can verify the network connectivity and handover between the assigned access points. Figure 11 shows that when Car1 crosses the ap1 coverage range, it automatically connects to ap2. Similarly, Train1 connects to access point ap4 automatically after crossing the coverage range of access point ap3. In Figure 11, we can see that after crossing the previously connected access points, Car1 and Train1 are disconnected, but there is no packet loss, and they are automatically connected to the nearest access point again.

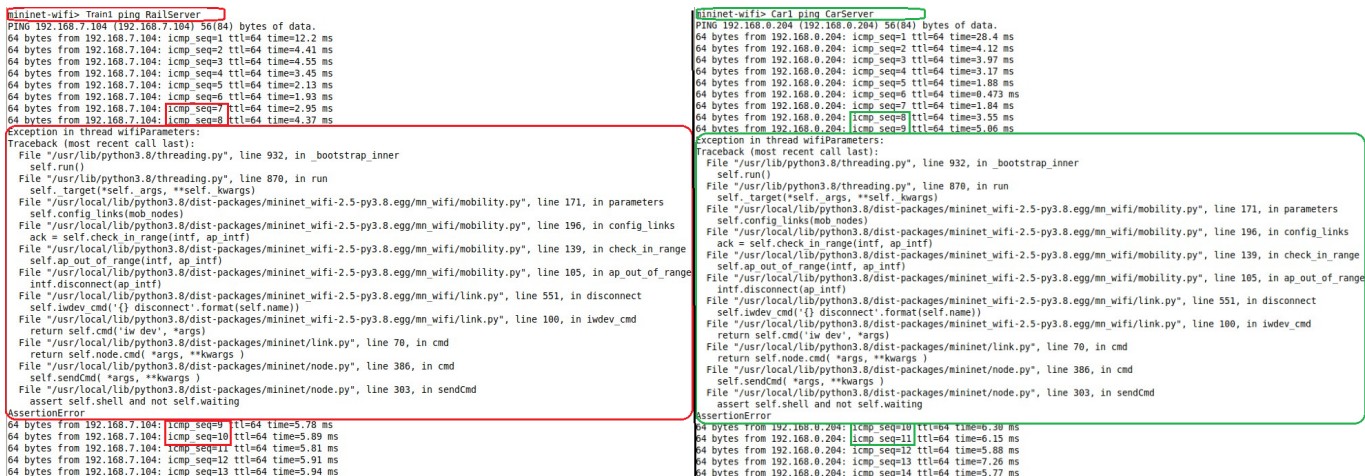

**Figure 11.** Checking connectivity during moving.

To verify the handover and moving functionality of nodes, command *Car1 iw dev Car1-wlan0 link* is executed for Car1, and for Train1 command *Train1 iw dev Train1-wlan0 link* is executed before and after the moving of nodes. Figure 12 shows that before the handover, Car1 is connected to access point ap1, and after handover, it is connected to access point ap2. Figure 13 shows that before moving, Train1 is connected to access point ap3, and after moving, it is connected to access point ap4. Figure 14 shows the topology after the movement of nodes Car1 and Train1, and it also shows that Car1 is now connected to access point ap2 and that Train1 is connected to ap4.

```
mininet-wifi> Car1 iw dev Car1-wlan0 link     mininet-wifi> Car1 iw dev Car1-wlan0 link
Connected to 02:00:00:00:08:00 (on Car1-wlan0) Connected to 02:00:00:00:09:00 (on Car1-wlan0)
        SSID: ssid-ap1                                 SSID: ssid-ap2
        freq: 5180                                    freq: 5200
        RX: 19167 bytes (409 packets)                 RX: 388479 bytes (8306 packets)
        TX: 2236 bytes (21 packets)                   TX: 46700 bytes (413 packets)
        signal: -27 dBm                               signal: -27 dBm
        rx bitrate: 54.0 MBit/s                       rx bitrate: 54.0 MBit/s
        tx bitrate: 54.0 MBit/s                       tx bitrate: 54.0 MBit/s

        bss flags:      short-slot-time               bss flags:      short-slot-time
        dtim period:    2                             dtim period:    2
        beacon int:     100                           beacon int:     100
                a. Before Handover                            b. After Handover
```

**Figure 12.** Connected access point for Car1 before and after handover/moving.

```
mininet-wifi> Train1 iw dev  Train1 -wlan0 link   mininet-wifi> Train1 iw dev  Train1 -wlan0 link
Connected to 02:00:00:00:0a:00 (on  Train1-wlan0) Connected to 02:00:00:00:0b:00 (on  Train1-wlan0)
        SSID: ssid-ap3                                SSID: ssid-ap4
        freq: 5180                                   freq: 5200
        RX: 33665 bytes (707 packets)                RX: 6081 bytes (129 packets)
        TX: 4444 bytes (41 packets)                  TX: 960 bytes (10 packets)
        signal: -27 dBm                              signal: -27 dBm
        rx bitrate: 54.0 MBit/s                      rx bitrate: 48.0 MBit/s
        tx bitrate: 54.0 MBit/s                      tx bitrate: 36.0 MBit/s

        bss flags:      short-slot-time              bss flags:      short-slot-time
        dtim period:    2                            dtim period:    2
        beacon int:     100                          beacon int:     100
                a. Before Handover                           b. After Handover
```

**Figure 13.** Connected access point for Train1 before and after handover/moving.

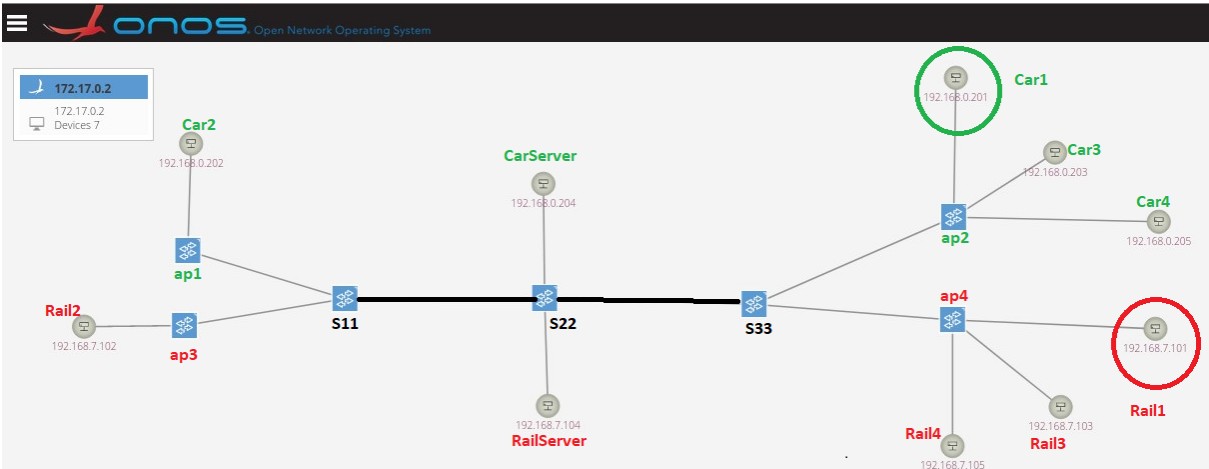

**Figure 14.** Handover scenario for S2(5/6)1 topology: ONOS screenshot.

Figure 4 shows the position of nodes and access points defined for scenario S2(5/6)1 before the moving scenario, and Figure 15 shows the positions of nodes and access points after moving and handover. After comparing these two graphs, we can conclude that Mininet–WiFi has the potential to emulate moving and handover scenarios for railway and road coexistence environments. There is some delay between the handover. This delay is because of the network joining process carried out by the nodes/stations and during this process; no data loss is recorded.

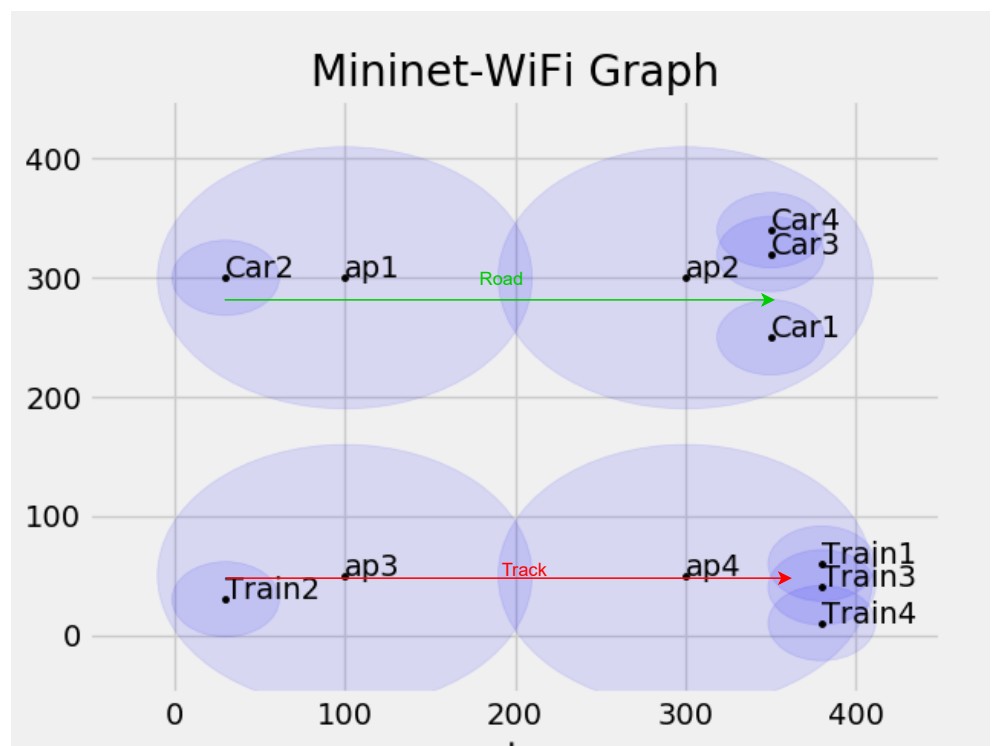

**Figure 15.** Hosts and access points after handover and moving: Mininet–WiFi graph.

### 9.2. Level Crossing

To manifest the level crossing scenario for railway and road coexistence environments, the S4(5/6)4 scenario is selected, which is presented in Section 6.2 of this paper. In this scenario, railway tracks are perpendicular to roads, as shown in Figure 7. We know that at a level crossing, when a Train/Tram is coming, Cars or other vehicles should have to

stop for some time and move when the train/tram has passed. It is one of the challenges for us to develop this scenario using the Mininet–WiFi and ONOS SDN controller. As we described in Section 9.1, the Mininet–WiFi Python API has a mobility method. Using this method, the user can start and stop the mobility of the nodes, but it does not have a pause and restart moving node functionality. To replicate the level crossing scenario, Car1 should stop its mobility when Train1 is moving at the level crossing point. To achieve this scenario, the *replayingMobility()* function is used, and the location (x- and y-coordinates) of nodes are saved in .dat files; these locations are provided using the function *get_trace()*, as shown in Listing 8.

**Listing 8.** Defining Moving Functionality of Hosts.

**net.isReplaying = True**

```
    path = os.path.dirname(os.path.abspath(__file__))
        + '/replayingMobility/'
    get_trace(Car1, '{}node1.dat'.format(path))
    get_trace(Car2, '{}node2.dat'.format(path))
    get_trace(Car3, '{}node3.dat'.format(path))
    get_trace(Train1, '{}node4.dat'.format(path))
    get_trace(Train2, '{}node5.dat'.format(path))
    get_trace(Train3, '{}node6.dat'.format(path))
```

The file node1.dat contains the location coordinates for the route of Car1 in virtual space. Figure 6 shows the initial location of Car1 and Train1. Car1's initial location is x = 10, y = 70, and after moving, the stop location x = 450, y = 70 is assigned. Train1's initial location is x = 130, y = 180, and the stopping location x = 130, y = −35 is assigned. To achieve the pause functionality for Car1, the coordinate x = 90, y = 70, where Car1 has to pause, is duplicated multiple times in the node1.dat file to reserve some time so that Train1 can pass the level crossing, as shown in Figure 16. Therefore, the *get_trace()* function makes Car1 pause at location x = 90, y = 70, and in that pause time, Train1 passes the level crossing; after that, Car1 starts moving again along the assigned route. Figure 17 shows that Car1 is paused at location x = 90, y = 60 and starts moving again when Train1 passed the level crossing.

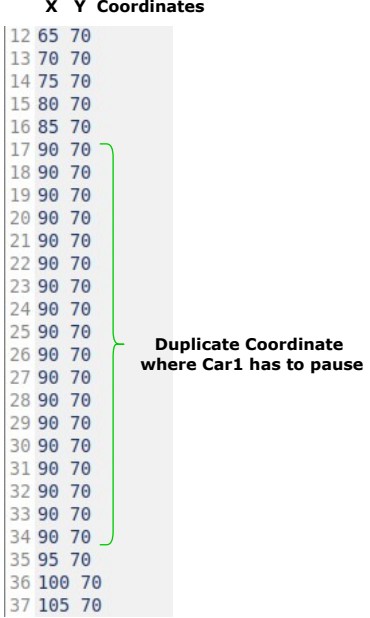

**Figure 16.** Car1: node1.dat file screenshot.

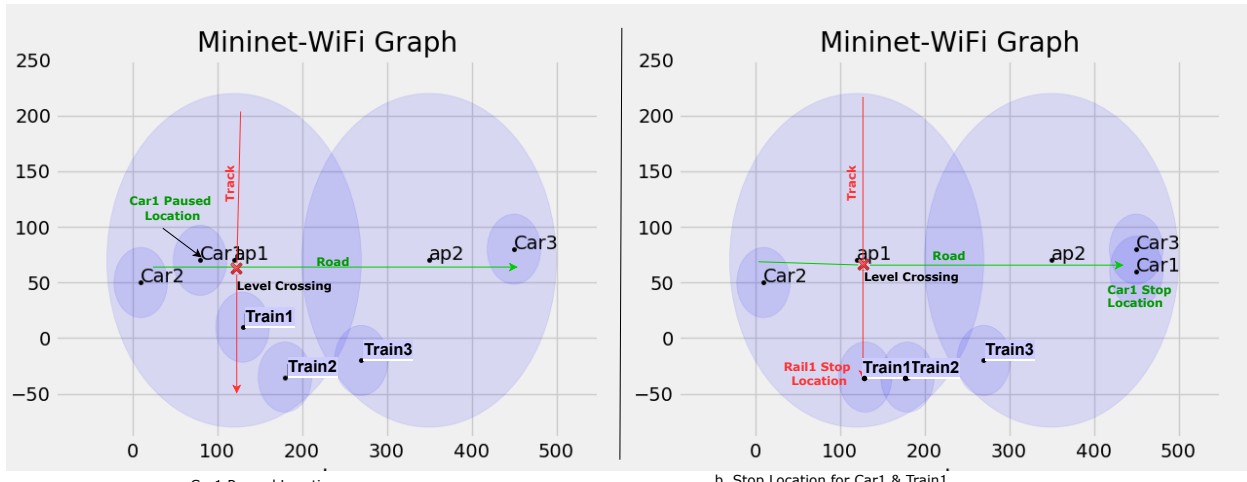

**Figure 17.** Level crossing scenario.

### 9.3. Reachability Test and Data Traffic Differentiation

To demonstrate the data traffic differentiation and network reachability test, scenario S2(5/6)1 is selected. In this scenario, the network topology has eight nodes, Car1, Car2, Car3, Car4, Train1, Train2, Train3, and Train4, representing cars and trains, respectively. It also has two hosts, CarServer and RailServer, assigned as service servers for roads and railways, respectively, as shown in the network topology in Figure 5.

The developed SDN application has the ability to differentiate the data traffic based on the assigned VLAN ID, and this test is carried out to validate the data traffic differentiation of the developed application. To carry out this test, the ping command is used to check the connectivity. Nodes Car1, Car2, Car3, Car4, and CarServer should be able to ping, send, and receive a data packet to/from each other but should not be able to communicate with trains and RailServer. In the same manner, Train1, Train2, Train3, Train4, and RailServer should be able to ping, send, and receive a data packet to/from each other but should not be able to communicate with cars and CarServer.

Figure 18 shows the connectivity test for node Car1. We can see that Car1 is able to ping other cars and CarServer, but it is not able to communicate with rails and RailServer. Similarly, Figure 19 shows the connectivity test for node Train1: it is able to communicate with other rails and RailServer but it is not able to communicate with cars and CarServer. This test is carried out for each and every node and host associated with network topology S2(5/6)1, and the outcome of this test is presented in Table 6, which shows that cars are able to communicate only with other cars and with the assigned road service server, i.e., CarServer, and trains are able to communicate only with other trains and with the assigned railway service server, i.e., RailServer.

**Table 6.** Reachability test.

| Src/Dst | Car1 | Car2 | Car3 | Car4 | CarServer | Train1 | Train2 | Train3 | Train4 | RailServer |
|---|---|---|---|---|---|---|---|---|---|---|
| Car1 | ✓ | ✓ | ✓ | ✓ | ✓ | X | X | X | X | X |
| Car2 | ✓ | ✓ | ✓ | ✓ | ✓ | X | X | X | X | X |
| Car3 | ✓ | ✓ | ✓ | ✓ | ✓ | X | X | X | X | X |
| Car4 | ✓ | ✓ | ✓ | ✓ | ✓ | X | X | X | X | X |
| CarServer | ✓ | ✓ | ✓ | ✓ | ✓ | X | X | X | X | X |
| Train1 | X | X | X | X | X | ✓ | ✓ | ✓ | ✓ | ✓ |
| Train2 | X | X | X | X | X | ✓ | ✓ | ✓ | ✓ | ✓ |
| Train3 | X | X | X | X | X | ✓ | ✓ | ✓ | ✓ | ✓ |
| Train4 | X | X | X | X | X | ✓ | ✓ | ✓ | ✓ | ✓ |
| RailServer | X | X | X | X | X | ✓ | ✓ | ✓ | ✓ | ✓ |

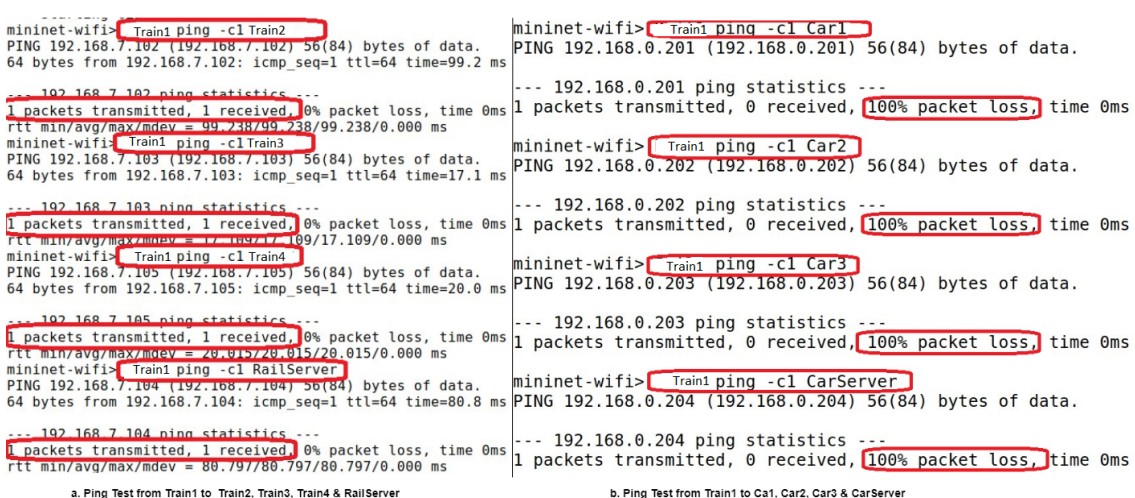

**Figure 18.** Reachability test for cars.

**Figure 19.** Reachability test for trains.

Figures 20 and 21 show the entries introduced by the developed SDN application on the switch S22. It shows that the developed SDN application is capable of differentiating the data packets from cars and trains by assigning a VLAN ID 4 to data packets to/from cars and CarServer and VLAN ID 3 to data packets to/from trains and RailServer. Figures 22 and 23 are the screenshots taken from the SDN application's log and show that the application restricts the communication between road entities and railway entities by maintaining connectivity and data traffic differentiation.

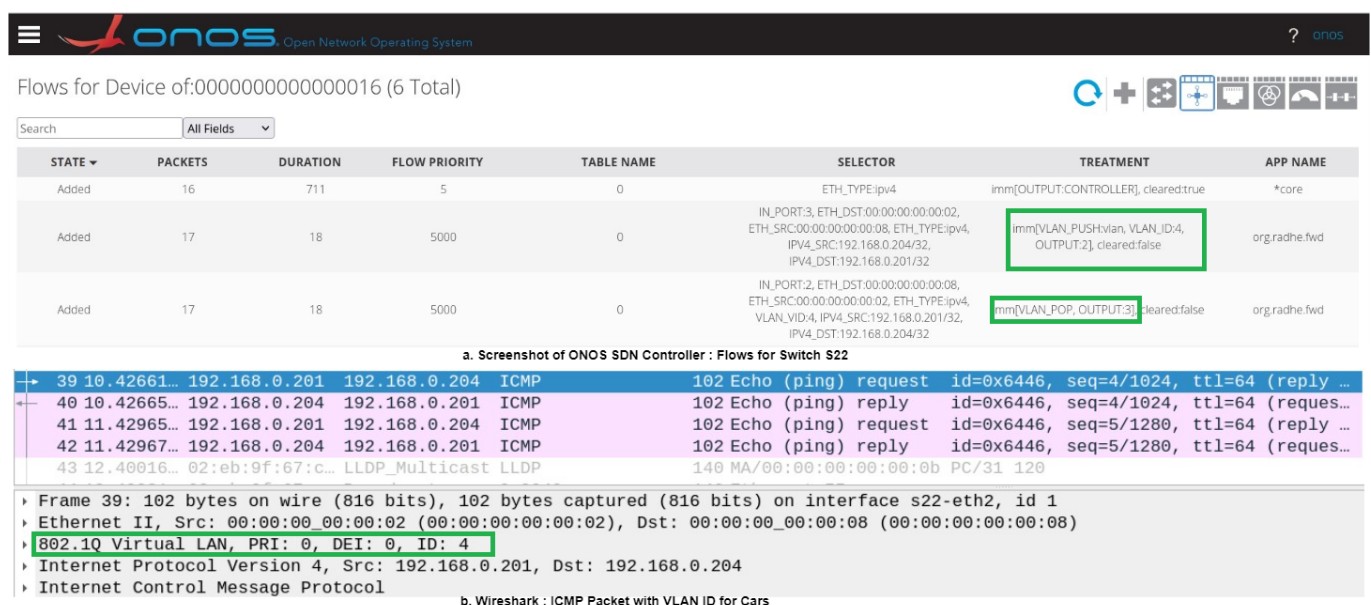

**Figure 20.** Flow entries for switch S22: VLAN ID assignment to car's data packet.

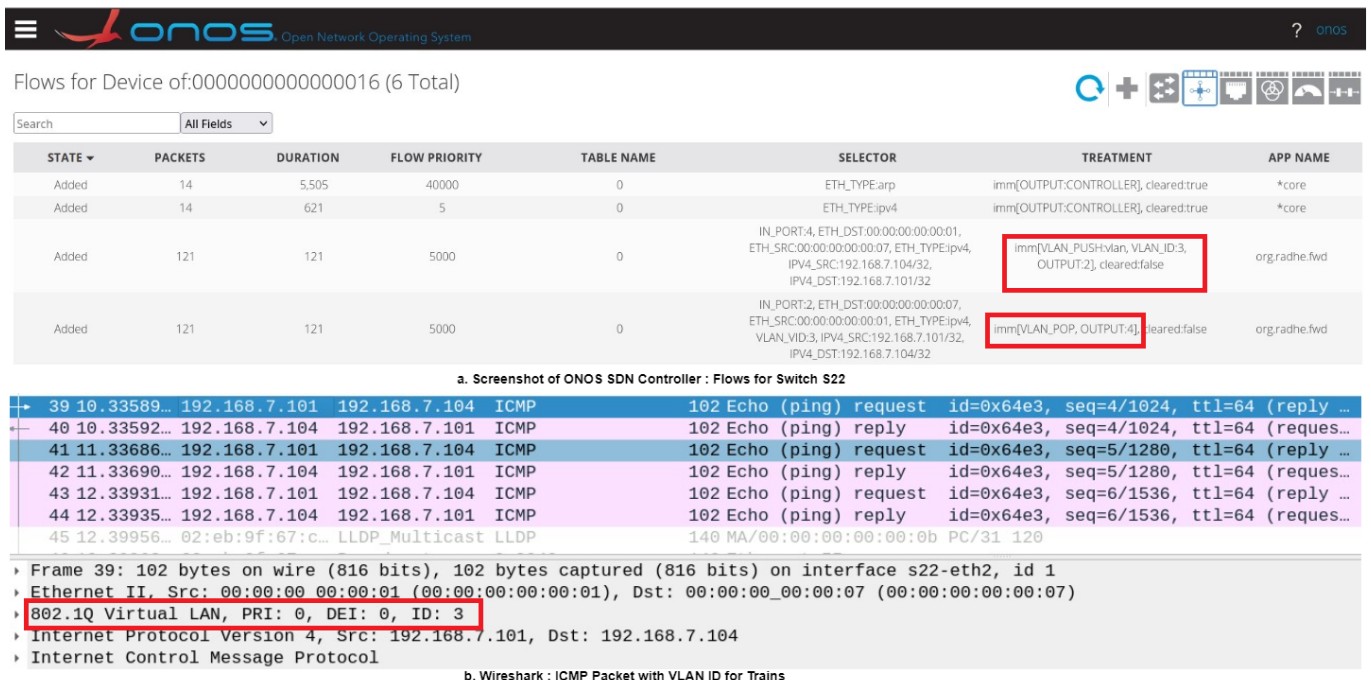

**Figure 21.** Flow entries for switch S22: VLAN ID assignment to train's data packet.

```
14:30:03.420 INFO [AppComponent]  ips not in same n/w or drop the packet:, srcip:192.168.0.201, descip:192.168.7.101
14:30:19.132 INFO [AppComponent]  ips not in same n/w or drop the packet:, srcip:192.168.0.201, descip:192.168.7.102
14:30:32.962 INFO [AppComponent]  ips not in same n/w or drop the packet:, srcip:192.168.0.201, descip:192.168.7.103
14:30:47.971 INFO [AppComponent]  ips not in same n/w or drop the packet:, srcip:192.168.0.201, descip:192.168.7.104
                                                                            Car1             Trains & RailServer
```

**Figure 22.** Screenshot of SDN application log: ping from Car1 to Train1, Train2, Train3, and RailServer.

```
14:21:53.116 INFO [AppComponent]  ips not in same n/w or drop the packet:, srcip:192.168.7.101, descip:192.168.0.201
14:22:07.893 INFO [AppComponent]  ips not in same n/w or drop the packet:, srcip:192.168.7.101, descip:192.168.0.202
14:22:21.911 INFO [AppComponent]  ips not in same n/w or drop the packet:, srcip:192.168.7.101, descip:192.168.0.203
14:22:39.800 INFO [AppComponent]  ips not in same n/w or drop the packet:, srcip:192.168.7.101, descip:192.168.0.204
                                                                            Train1           Cars & CarServer
```

**Figure 23.** Screenshot of SDN application log: ping from Train1 to Car1,Car2, Car3, and CarServer.

### 9.4. TCP and UDP Data Transmission

The objective of this test is to demonstrate the standard data communication between cars to CarServer and trains to RailServer. Figure 24 shows UDP data packet transmission and Figure 25 shows TCP data packet transmission from Train1 to RailServer. In this case, Train1 is acting as the client, and RailServer is configured as a listening server.

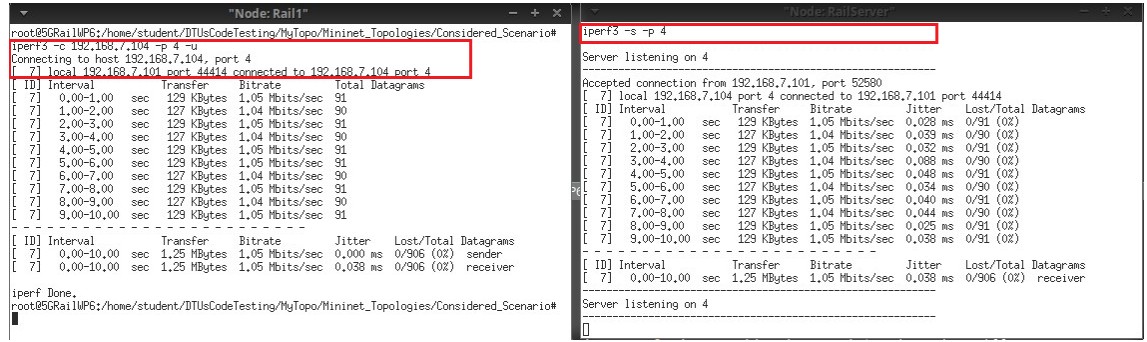

**Figure 24.** UDP data transmission from Train1 to RailServer.

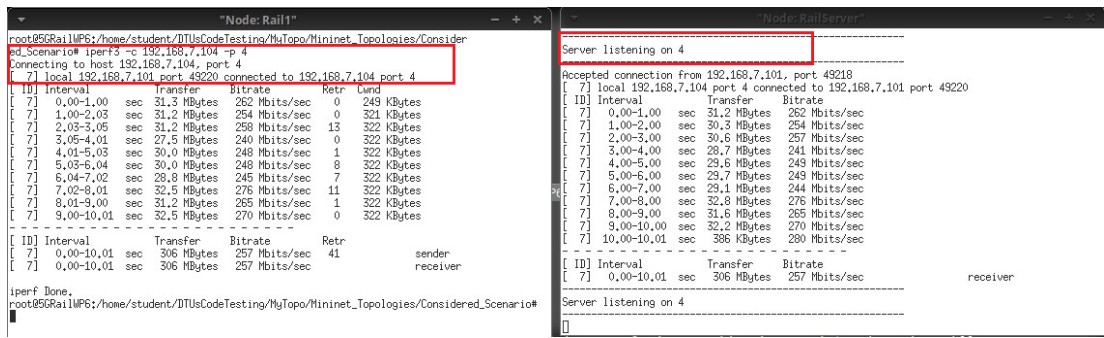

**Figure 25.** TCP data transmission from Train1 to RailServer.

### 9.5. Link Capacity Test

This test is carried out using the iperf tool to measure the bandwidth between two network links. To measure the bandwidth, the *iperf <Host1> <Host2>* command is used. Figure 26 shows the link capacity measurement between Car1 and CarServer and Train1 and RailServer. The achieved bandwidth measurement shows that it is adequate to send and receive messages, voice, and video data for coexistence scenarios for roads and railways.

```
*** Starting CLI:
mininet-wifi> iperf  Train1  RailServer
*** Iperf: testing TCP bandwidth between  Train1  and RailServer
*** Results: ['259 Mbits/sec', '259 Mbits/sec']
mininet-wifi> iperf Car1 CarServer
*** Iperf: testing TCP bandwidth between Car1 and CarServer
*** Results: ['237 Mbits/sec', '239 Mbits/sec']
mininet-wifi>
```

**Figure 26.** Link capacity test.

### 9.6. Latency Test and Network Jitter Test

The MTR tool is used to measure losses, latency, and network jitter. To conduct the latency test, 100 UDP and TCP data packets are sent from Car1 to CarServer and Train1 to RailServer. This is carried out by opening the xterm window for nodes Car1 and Train1 using the command **xterm Car1 Train1** from the Mininet–WiFi terminal. To measure the latency from Car1, the command *mtr -r -n -c 100 192.168.0.204 -u -P 3* is used, where 192.168.0.204 is the IP address of CarServer and is connected to port 3 of network switch

S22. Similarly, the command *mtr -r -n -c 100 192.168.7.104 -u -P 4* is executed from Train1 to measure the latency while sending 100 UDP data packets to RailServer. The IP address of the assigned RailServer is 192.168.7.104, and it is connected to port 4 of network switch S22. The latency for this network topology is in the range of 4.7 to 5.7 milliseconds. Figures 27 and 28 show the latency test for the selected network topology.

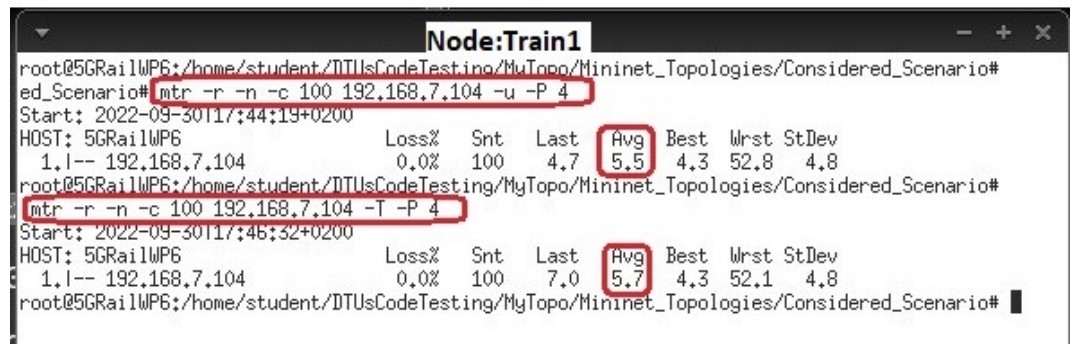

**Figure 27.** Latency test from Car1.

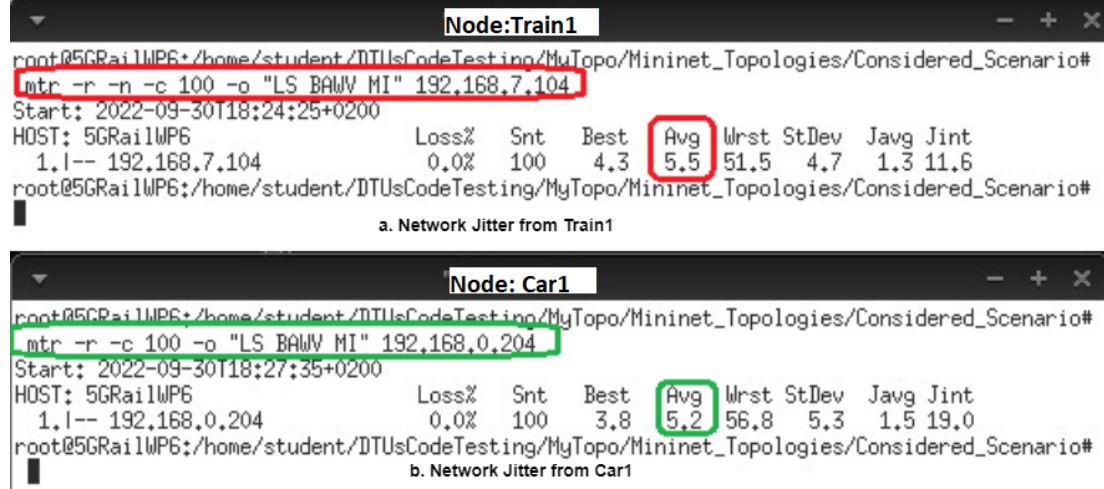

**Figure 28.** Latency test from Train1.

To measure the jitter of the network, the command *mtr -r -n -c 100 -o "LS BAWV MI" 192.168.7.104* is executed from the terminal of node Train1, and the command *mtr -r -n -c 100 -o "LS BAWV MI" 192.168.0.204* is executed from the terminal of node Car1. The information about commands and parameters used is given in Table 5. Figure 29 shows that the jitter of the network is in the range of 5.2 to 5.5 milliseconds.

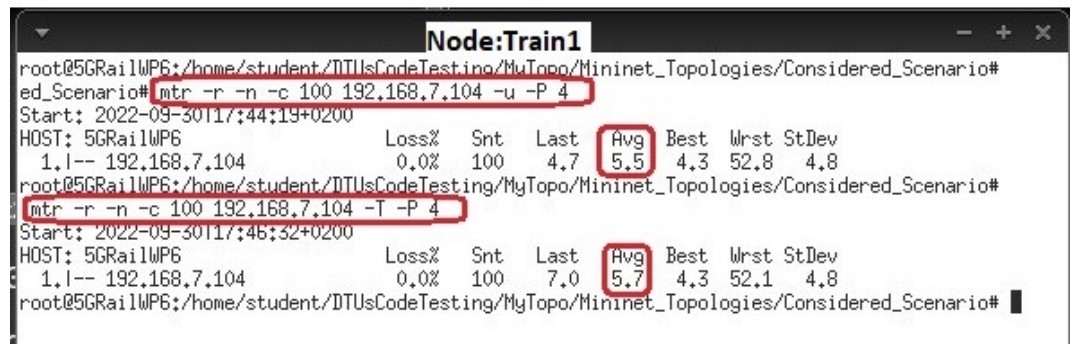

**Figure 29.** Network jitter test.

*9.7. Sending a Critical Message to the Assigned Server*

To send a critical message or information, the Scapy tool is used for the considered scenarios. Using this tool, a user-defined message is sent from any node/station to the assigned service server.

Figure 30 shows that an ICMP data packet is sent with a message *"Emergency Msg: Engine Failure"* from node Train1 to RailServer for demonstration purposes in the selected scenario. This data packet can be sent using the Scapy Python API and by writing a Python script. Figure 31 shows the data packet captured using the Wireshark tool.

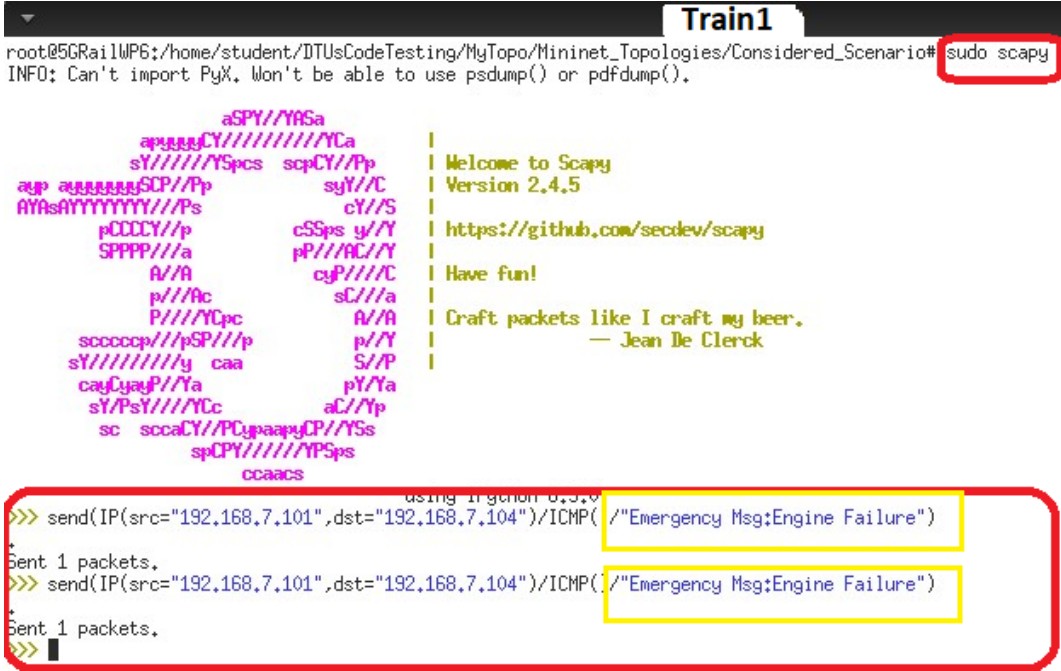

**Figure 30.** Scapy: packet creation.

**Figure 31.** Wireshark: Scapy packet with a message.

The Python scripts given below represent sending critical data from node Train1 to RailServer using ICMP protocol.

**Listing 9.** Sending a Critical Message Using Scapy with Python.

```python
#! /usr/bin/env python
# The following line will import all Scapy modules
from scapy.all import *
i = 1
while i < 50:
    send(IP(src=''192.168.7.101'' ,dst=''192.168.7.104'')/ICMP()
    /''Emergency_Msg:Engine_Failure'')

    i += 1
    print(''Train1_is_sending_msg_to_RailServer'')
```

*9.8. Video Transmission Test*

To demonstrate the video transmission for railway and road coexistence scenarios, VLC player is used. A detailed procedure is elaborated below with figures to transmit a video file from node Train1 to RailServer to manifest video communication or critical video communication. This is implemented by opening the xterm window for Train1 and Railserver using the command *xterm Train1 Railserver* and running the command *vlc-wrapper & from both instances of xterm. Figure 32 shows the initial window after running the command, where Train1 is configured to transmit the video to the node RailServer.

- **VLC player configuration at Train1:** Train1 is acting as the video transmitter. Click on the option "Media", select the option "Stream", and add the video file as shown in Figure 33. After selecting the option "Stream", a new window will pop up that provides the information about the selected video file; select the streaming method as "RTP/MPEP Transport Stream", as shown in Figure 34.

  Add the destination host's IP address. Here, the destination IP address is 192.168.7.104, which belongs to RailServer. In the next window, as shown in Figure 35, select the transcoding option "video-H.256 + MP3(MP4)"; other options can be selected based on the user's transmission video format.

  Now, select the option "Stream", as shown in Figure 36 and start streaming the video from Train1 to RailServer.

- **VLC player configuration at RailServer:** From "Media", select option "Open Network Stream", and a new window will pop up; add the network URL as "rtp://@:5004", as shown in Figure 37.

Figure 38 shows that Train1 is streaming the video and RailServer is receiving the video. Therefore, it can be concluded that by using the VLC player, a user can demonstrate video transmission from one host to another host in a railway and road coexistence scenario.

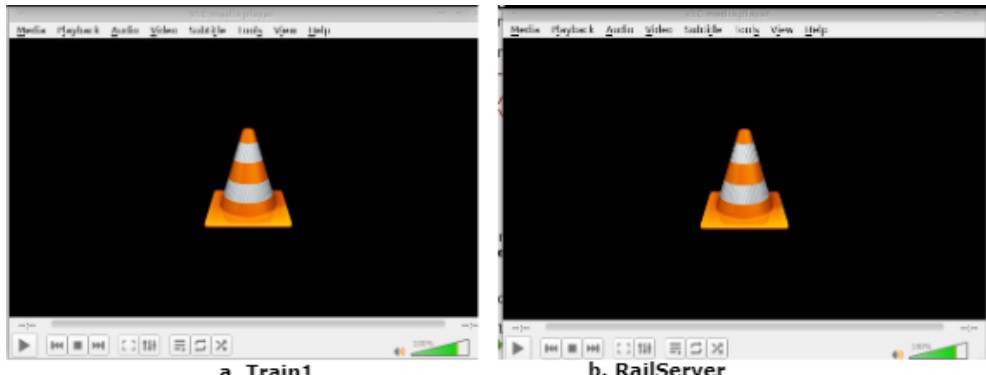

**Figure 32.** VLC player execution At Train1 and RailServer.

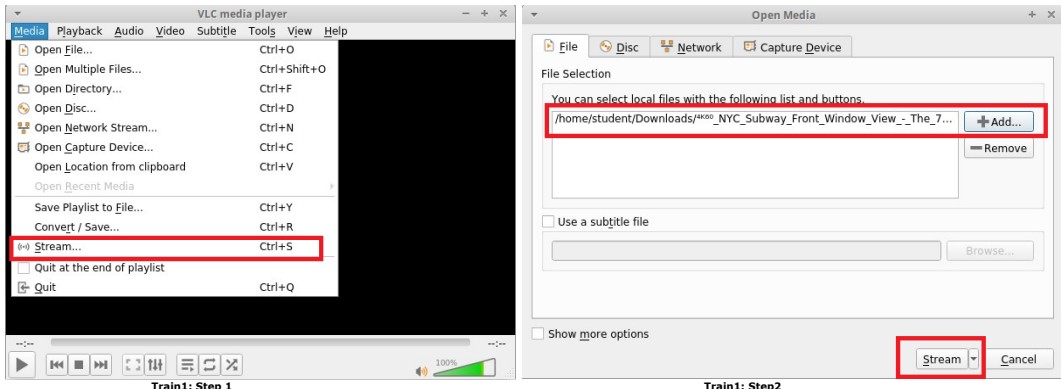

**Figure 33.** Train1: adding the video file to transmit.

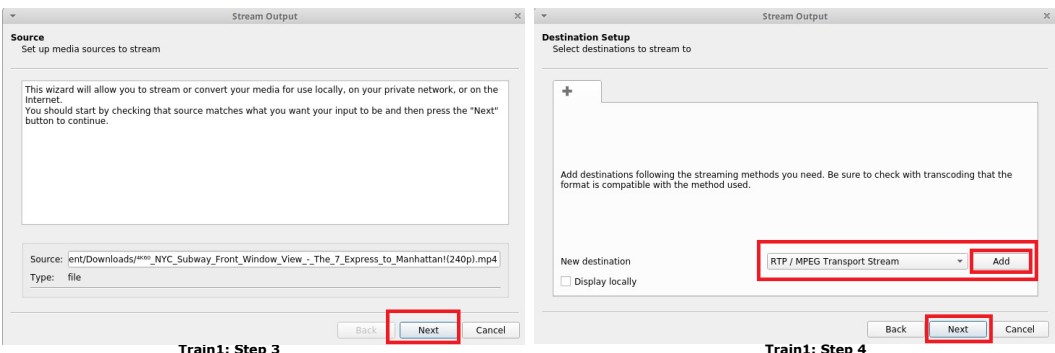

**Figure 34.** Train1: selecting the streaming method.

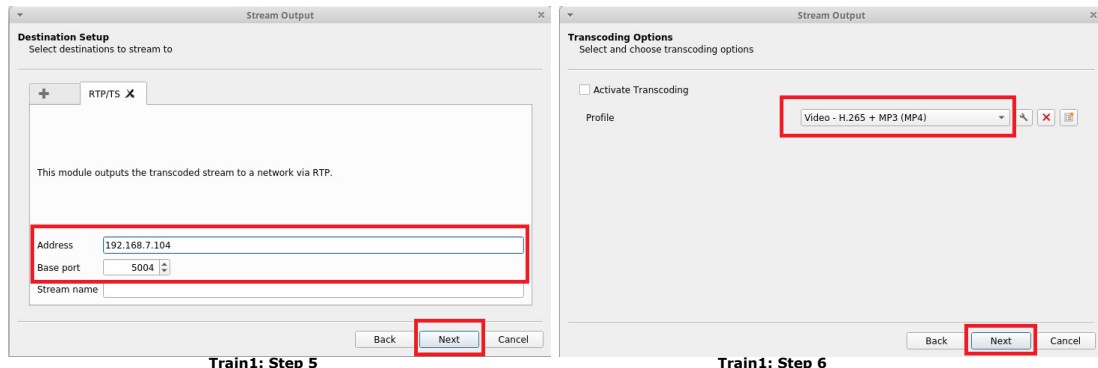

**Figure 35.** Train1: adding the destination IP address of RailServer.

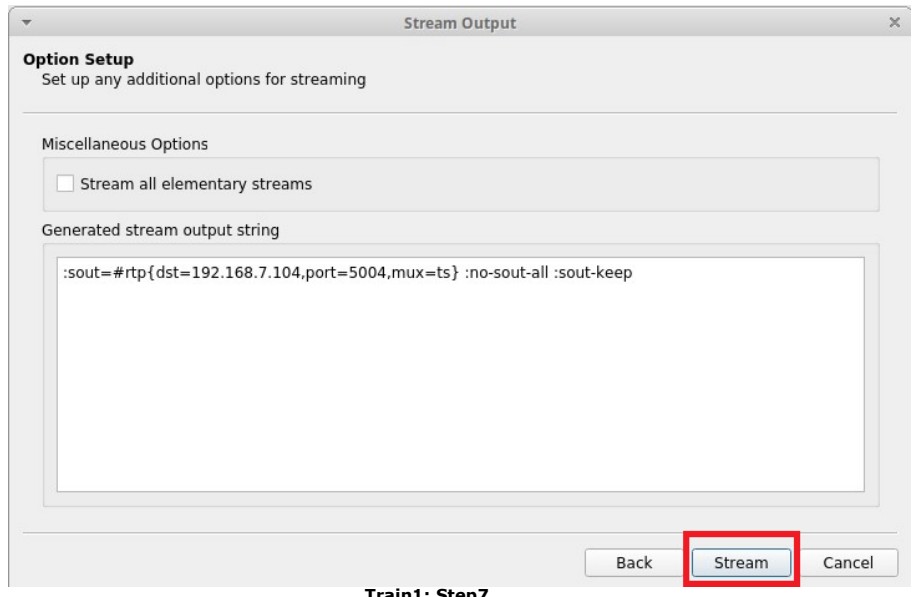

**Figure 36.** Train1: stream the video.

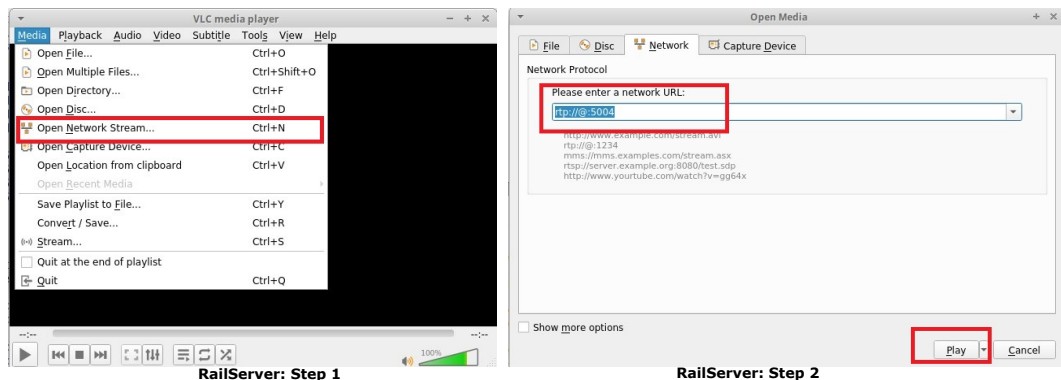

**Figure 37.** Train1: stream the video.

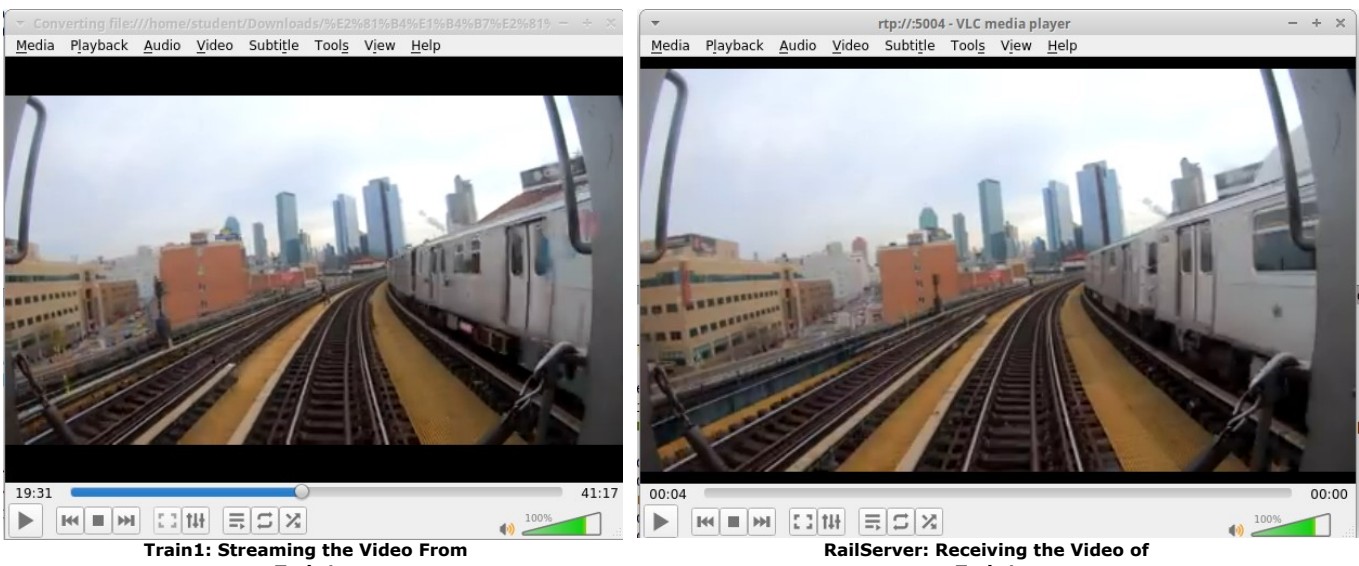

**Figure 38.** Video data transmission and reception [39].

## 10. SUMO Integration

In this section, a detailed description of SUMO integration with Mininet–WiFi is given to showcase the graphical representation of railway and road coexistence scenarios. Figure 39 shows the steps to generate the SUMO map from open street map and integration with Mininet–WiFi.

- **Map Generation:** To show the coexistence scenario of railways and roads in a pictorial form, user can design the desired map. A detailed tutorial is presented in [40]. For this practical work, we download the desired Google map file. To generate the desired emulation map on Google street map, run the command *python3 /usr/share/sumo/-tools/osmWebWizard.py* from the folder where you want to download the map files. After running the command, the "OSM Web Wizard for SUMO" page will open. Go to the "Generate Scenario" option given on the right side of the page and select the map location by providing the city or place name or by providing the GPS coordinates of the region. Figure 40 shows the OSM Web Wizard home page. To demonstrate SUMO integration, we select the place "Puente De Santiago" since it has tracks parallel to roads. The map area is selected using the option "Select Area". Figure 41 shows the selected map area and selected parameters to generate the vehicle traffic. Using the parameter "Through Traffic Factor", the user can define the number of vehicles that depart and arrive at the selected simulation boundary area.

After selecting the required parameters, by clicking on the "Generate Scenario" option, all the map files will be downloaded to the selected folder. After completion of map file generation, the user will get the messages shown in Figure 42. If all the steps are correct and the intended map files are downloaded, a new window will pop up and start running the simulation, as shown in Figure 43.

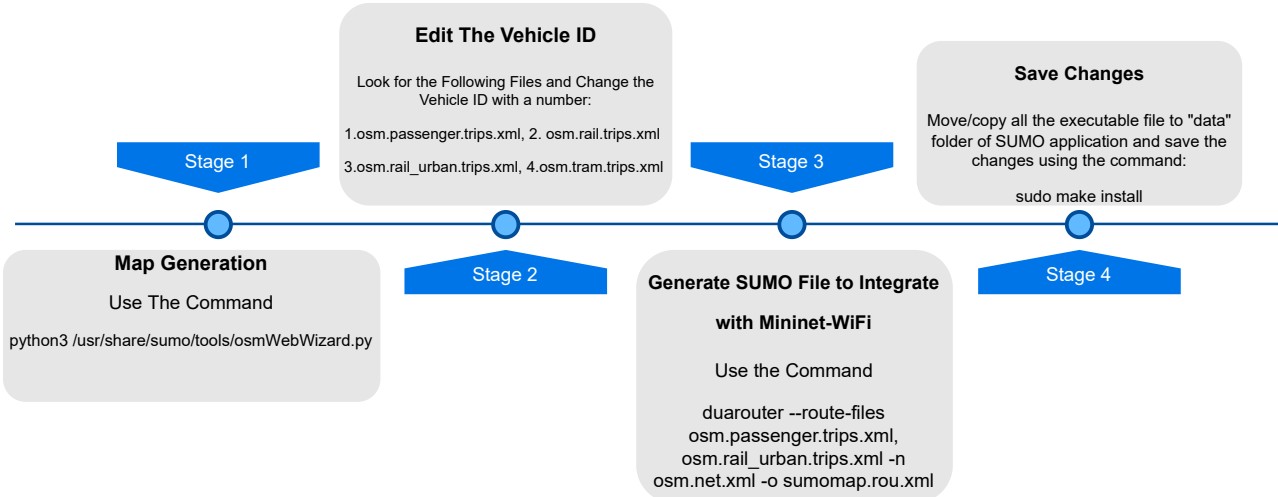

**Figure 39.** SUMO integration steps.

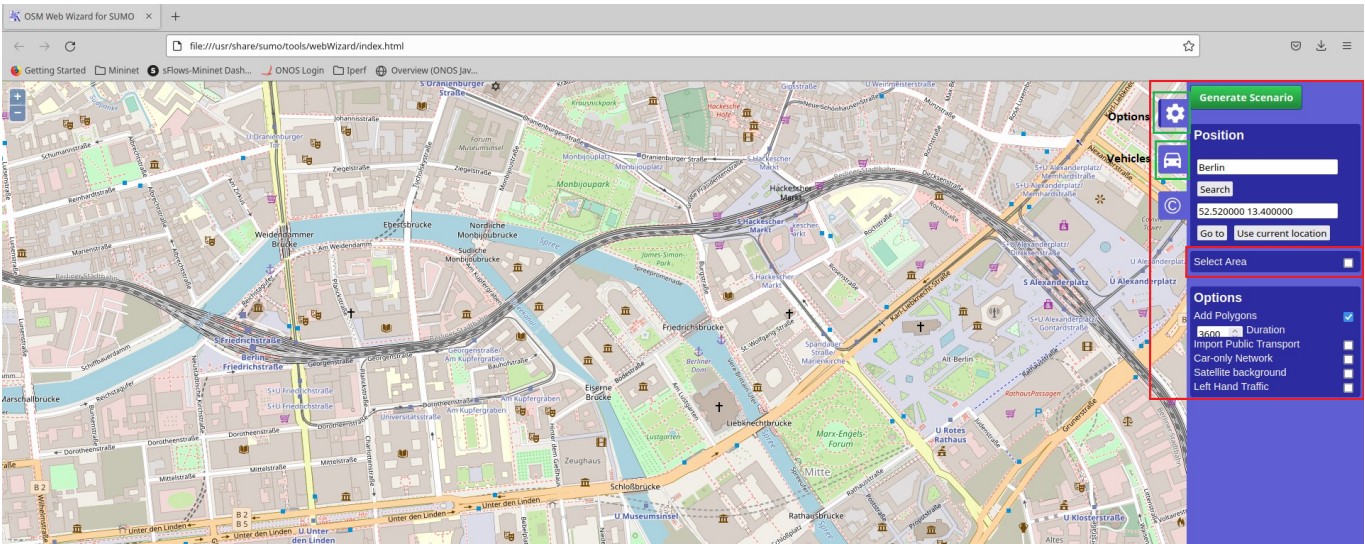

**Figure 40.** OSM Web Wizard home page.

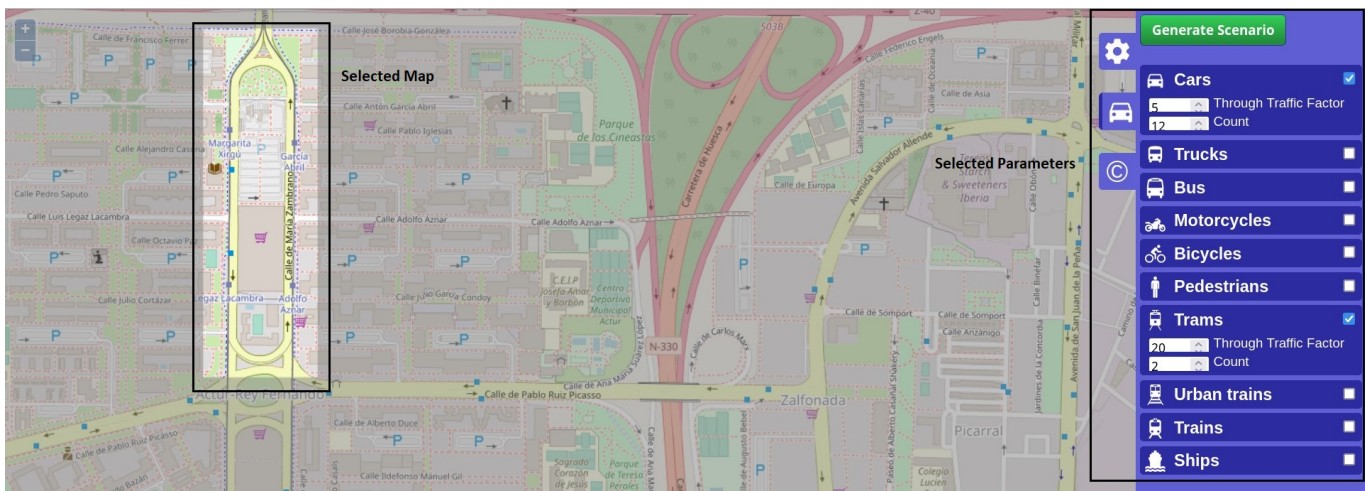

**Figure 41.** Selected map: Puente de Santiago.

```
student@5GRailWP6:~/MAPforSUMO$ python3 /usr/share/sumo/tools/osmWebWizard.py
Building scenario in '/home/student/MAPforSUMO/2022-07-13-14-26-13'
Downloading map data
200 OK
Converting map data
Written configuration to 'osm.netccfg'
Loading configuration ... done.
Parsing types from '/usr/share/sumo/data/typemap/osmNetconvert.typ.xml' ... done.
Parsing nodes from osm-file 'osm_bbox.osm.xml' ... done.
Parsing edges from osm-file 'osm_bbox.osm.xml' ... done.
Removing duplicate edges ... done.
Warning: Discarding unknown compound 'cycleway.track' in type 'cycleway.track|highway.tertiary' (first occurence for edge '3101253').
Warning: Discarding unusable type 'waterway.river' (first occurence for edge '23883377').
Warning: Discarding unknown compound 'cycleway.shared_lane' in type 'cycleway.shared_lane|highway.residential' (first occurence for edge '25417691').
Warning: Discarding unusable type 'waterway.canal' (first occurence for edge '25637137').
Warning: Discarding unknown compound 'usage.branch' in type 'railway.light_rail|usage.branch' (first occurence for edge '34670067').
Warning: Discarding unusable type 'railway.platform' (first occurence for edge '121041773').
Warning: Discarding unknown compound 'cycleway.lane' in type 'cycleway.lane|highway.tertiary' (first occurence for edge '171328774').
Warning: Discarding unusable type 'route.ferry' (first occurence for edge '240704224').
Warning: Discarding unusable type 'waterway.stream' (first occurence for edge '262972999').
Warning: Discarding unknown compound 'cycleway.track' in type 'cycleway.track|highway.residential' (first occurence for edge '403904474').
Parsing relations from osm-file 'osm_bbox.osm.xml' ... done.
 Removed 5 traffic lights before loading plain-XML
Import done:
  779 nodes loaded.        >Map Files are Generated and Downloaded
  33 types loaded.
  1407 edges loaded.
Proj projection parameters used: '+proj=utm +zone=33 +ellps=WGS84 +datum=WGS84 +units=m +no_defs'.
Removing self-loops ... done (0ms).
Joining junction clusters ...
Warning: Reducing junction cluster 1301026983,2024780,2027668,2484127698,402612887,6953986825,6953986826,8447283388,8447283395,8811679734,8811679736,8811679738,9
Warning: Reducing junction cluster 3576199722,3576199733,3576199740,6750108975,6750120293,6750120298,6750120300,6750120310,6750120473,6750138592,67501
Warning: Reducing junction cluster 2027682,402608030,402608032 (parallel incoming -34722847#0,339889572#1).
Warning: Reducing junction cluster 2086883757,2086883770,653742583 (6 incoming edges).
Warning: Reducing junction cluster 2086883795,530811646,6749940366 (5 incoming edges).
done (7ms).
```

**Figure 42.** Map generation.

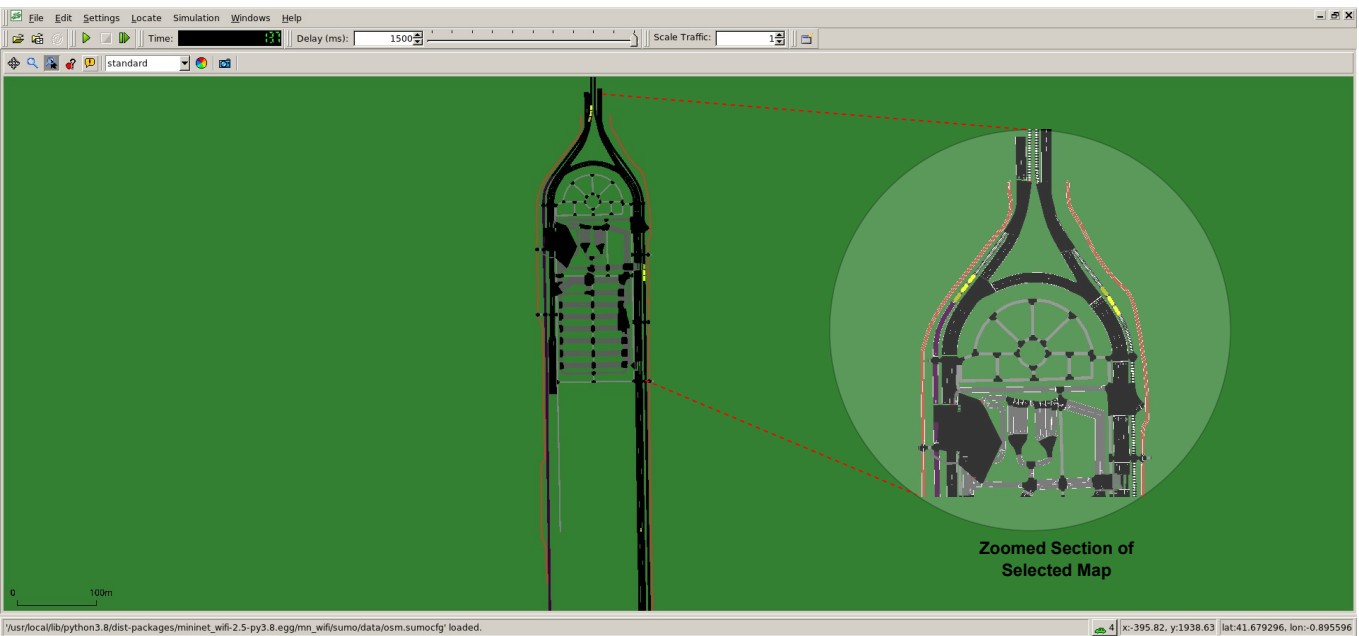

**Figure 43.** Simulation: Puente de Santiago

- **Edit the Vehicle Id, i.e., Car ID and Train/Tram ID:** To show the vehicles in SUMO and to interface with Mininet–WiFi, the selected vehicles/nodes should have different IDs. Go to the folder where map files are generated and look for the files that have the car and the tram. Look for the following emulation files:

  1. **osm.passenger.trips.xml**: contains the car ID;
  2. **osm.rail.trips.xml**: contains train ID (long route trains);
  3. **osm.rail_urban.trips.xml**: contains urban train ID;
  4. **osm.tram.trips.xml**: contains tram ID.

  In this selected map, a car and a tram are simulated as network nodes. Open the files *osm.passenger.trips.xml* and *osm.tram.trips.xml* and change the IDs with numerical numbers such as 1, 2, 3, etc.; the number should not be repeated; every entity should have a different number, as shown in Figure 44. After that, to build a single executable file, open a terminal in the folder where all these files are saved and run the command:

  > *duarouter - -route-files osm.passenger.trips.xml, osm.rail_urban.trips.xml -n osm.net.xml -o sumomap.rou.xml*

  where *sumomap.rou.xml* is the executable file. Figure 45 shows that the *sumomap.rou.xml* file is generated.

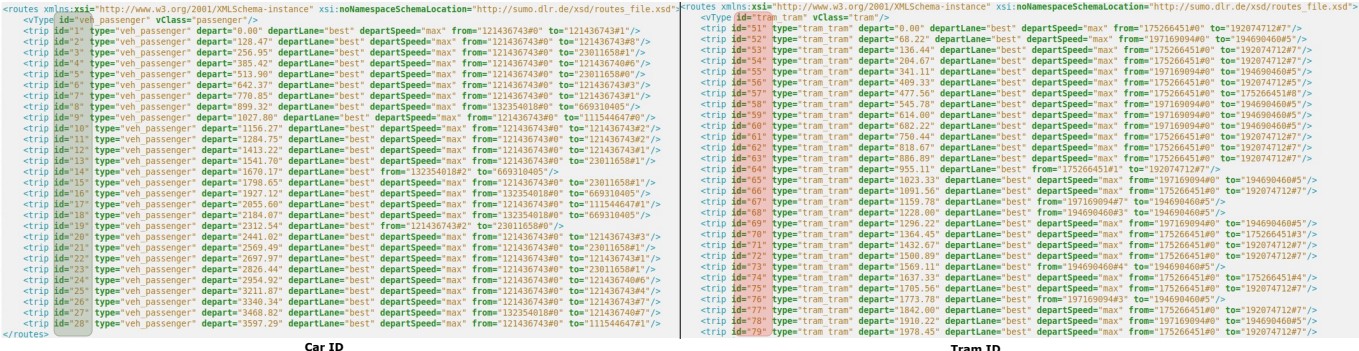

**Figure 44.** Change IDs of vehicles.

**Figure 45.** Change IDs of vehicles.

- **Edit the File *osm.sumocfg*:** After generating the map file *sumomap.rou.xml*, open the file *osm.sumocfg* and change the following information:

  <route-files value="sumomap.rou.xml"/>

- **Move or Copy the Map Files:** By default, Mininet–WiFi accesses the SUMO files from the "data" folder of the SUMO application. Therefore, after completing all the steps mentioned above, move or copy the map files in the /home/student/mininet-wifi/mn_wifi/
  sumo/data folder.

- **Save all Changes:** Now, go to the folder for Mininet–WiFi, open a terminal, and run the following command *sudo make install* to save all the changes done with SUMO files, as shown in Figure 46.

**Figure 46.** Save all the changes.

- **SUMO Map Integration with Selected Network Topology:** To generate the network topology integrated with SUMO maps, a Python script is used utilizing all the methods and functions described in Section 6.1. The user has to add the following line of code mentioned in List 10 to the network topology Python script. The command *net.useExternalProgram* is used to integrate the SUMO application with Mininet–WiFi. The entire Python script (*SUMO_Aug17.py*) for network topology creation with SUMO map integration is available at [26].

**Listing 10.** SUMO Integration with Mininet–WiFi.

```
info(''****Starting_network_and_connecting_to_traci'')
    info('Connecting_to_traci_sumo')

    net.useExternalProgram(program=sumo, port=8813,
```

```
config_file='osm.sumocfg',
extra_params=[''--start --delay 1500''])
```

After running the script *SUMO_Aug17.py*, network emulation with the SUMO map as shown in Figure 43 will be generated. Figure 47 shows the location of the assigned access points and the movement of cars and trams, where assigned access points are ap1 to ap17, C2 represents Car2, and T77 represents Tram77. When a car/tram moves from the coverage range of one access point to another, we sometimes observe that there is no automatic connection to the nearest access point. In that case, the connection can be established using the command *<node name> iw dev <node name>-wlan0 connect <SSID name>*.

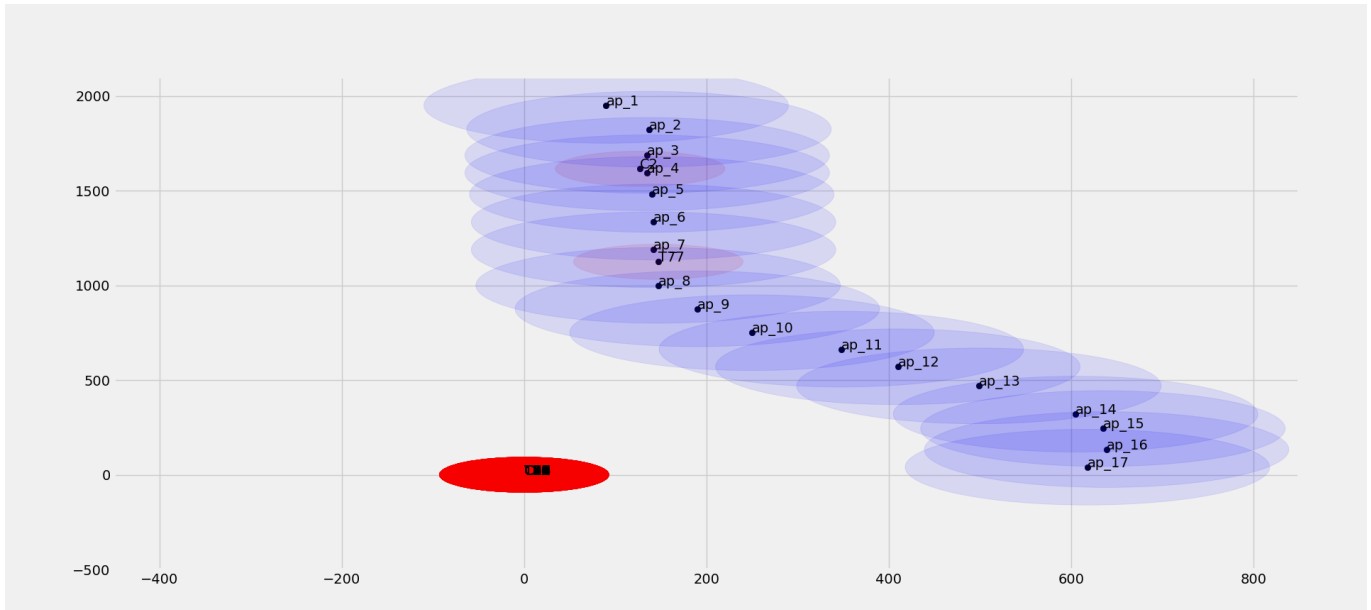

**Figure 47.** Movement of trams and cars on SUMO map in the range of assigned access points.

## 11. Conclusions

In this paper, we have provided a detailed tutorial to emulate the coexistence of railway and road services by sharing telecommunication infrastructure using the tools Mininet–WiFi, ONOS-SDN Controller, and SUMO. Based on the validation test carried out in Section 9, it can be concluded that by using Mininet–WiFi, a user can develop different network topologies with nodes having moving capabilities and wireless access points intended for railway and road coexistence scenarios. Therefore, moving hosts are able to replicate cars, rails, and trams in considered virtual space. As we have stated, Mininet–WiFi does not support 5G-based network emulation. In our previous work [16], we presented a network emulator "Emu5GNet". This emulation tool allows the emulation of 5G networks with complex applications. The developed ONOS SDN application is able to differentiate data traffic based on the VLAN tag. Along with this, the developed SDN application is able to handle the handover scenarios. The visualization tool SUMO is capable of representing the simulation of a considered scenario in a pictorial/graphical way. After executing the Mininet–WiFi network emulation file with SUMO, there are sometimes no automatic connections of nodes to the nearest WiFi access points when they move from one access point to another. This may be because of a coverage range problem where vehicles are moving. Since we have extracted the SUMO map from open street map, we do not know exactly the design and shape of the road. This is subjected to only SUMO integration. To mimic real data traffic for the considered coexistence scenarios for railways and roads, iperf3, Scapy, and VLC player are considered. Using iperf3, standard data communication is demonstrated by sending and receiving UDP and TCP packets between nodes. Using Scapy, messaging and critical data communication are manifested. Video transmission

from one network node to another node is demonstrated using VLC Player. The MTR tool is used to measure the network parameters of latency, packet loss, and jitter. Therefore, it can be concluded that the considered tools have the potential to emulate the considered scenarios for railway and road coexistence environments.

**Author Contributions:** Conceptualization, J.S., R.S., T.S., L.M. and M.B.; methodology, J.S. and R.S.; software, R.S. and J.S.; validation, J.S., R.S., T.S., L.M. and M.B.; formal analysis, R.S. and J.S.; investigation, R.S. and J.S.; resources, J.S., R.S., T.S., L.M. and M.B.; data curation, J.S., R.S., T.S., L.M. and M.B.; writing—original draft preparation, R.S. and J.S.; writing—review and editing, J.S., R.S., T.S., L.M. and M.B.; visualization, R.S. and J.S.; supervision, J.S.; project administration, J.S., L.M. and M.B.; funding acquisition, J.S., L.M. and M.B. All authors have read and agreed to the published version of the manuscript.

**Funding:** This empirical work is part of the project "5G for future RAILway mobile communication system" (5GRAIL). It is funded by the European Union's Horizon 2020 research and innovation program. The grant agreement number is 951725.

**Institutional Review Board Statement:** Not applicable.

**Informed Consent Statement:** Not applicable.

**Data Availability Statement:** Not applicable.

**Conflicts of Interest:** The authors declare no conflict of interest.

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
