# Peer review of "Coexistence of Railway and Road Services by Sharing Telecommunication Infrastructure Using SDN-Based Slicing: A Tutorial"

_2673-8732, doi:10.3390/network2040038_

Round 1

Reviewer 1 Report

The paper presents a detailed tutorial that demonstrates how to emulate the coexistence of railways and road services by using a variety of tools. Find my comments:

·       There is a lack of consistency in the motivation for this research, which can be clarified. As an example, the necessity of this research has not been adequately investigated. Furthermore, the problem statement that should be addressed by this study has not been identified. Consideration should be given to this issue.

·       Ref [13] has not been cited in the text.

·       I find this statement to be a bit vague: "From the research work mentioned above, it is apparent that narrow research work or emulation work is being conducted in the field of vehicular networks based on software-defined networks.". After reviewing existing studies, readers are looking for what is lacking in the literature.

·       A general observation is that the related works section is very weak and does not support the structure of the paper. Many other studies have not been analyzed in order to highlight the considered methodology for addressing the research gap (which has not been adequately addressed). Several studies have not considered the use of SDN-based slicing in railways.

·       Has any of the selected tools been used in similar applications in the literature?

·       Ideally, the contribution of the paper should be described in the form of an architecture system. A detailed description of each element shall be provided. The current state is very confusing. The contribution and proposed architecture should be elaborated in Section 6.

·       It is not justified to select scenarios based on a previous report by the same project ref[15].

·       The following sentence should be written in the related works: "Our previous work [40] presented a network emulator....". It should be part of the section on related works that aim to illustrate the gaps in the existing work.

·       There are a few typos, and grammatical errors, such as "SUMO is capable to represents the...".

·       Neither the limitations nor the advantages of the system have been considered. As an example, this sentence should take into account the advantages of the proposed system, since this paper focuses on implementation rather than only conceptual design "Therefore, it can be concluded that the considered tools have...".

·       Although it is mentioned that the "MTR tool is used to measure the parameters of the network...", these parameters are not discussed in the paper.

·       Overall, the paper demonstrates the implementation of the techniques, but not the justification behind their use, nor is the performance of the tools adequately described.

Author Response

Dear Sir,

Thanks a lot for such a detailed review. 

BR,

Radheshyam Singh

Reviewer 2 Report

Dear authors, see my comments below:

1. Please accurately express the relationship between SDN and slicing. How they can be adopted in the co-existence 836 of railways and road services;

2. Correct the typos and improve writing quality.

Author Response

(The authors gave the same response as above.)

Reviewer 3 Report

The paper provides a detailed tutorial to emulate the co-existence of railways and road services by sharing telecommunication infrastructure using the tools Mininet-WiFi, ONOS-SDN Controller, and SUMO. On the basis of relevant works, this paper considers the coexistence scenario and provides a richer application for tools they select. While it lacks novelty in terms of additional contributions. In addition, the following comments should be considered.  

1. How to understand that hosts which can move in the given direction at an assigned speed?Is that just mean end points, or it is realized by using virtual host in mininet?

2. What is the difference between railways and roads in the coexistence scenario, or the differences between coexistence and non-coexistence scenarios,  cause it is mentioned that compared with previous works, the paper is the first-one, while the key point has not been clearly reflected.

3. In Section ” Essential Parameters to Define the Scenario” 3.1B “Mobility Parameters ”, there are kinds of parameters, ex, highway,tram,urban train, that have not been displayed fully, does this tool design and implement all of them?

4. The keywords seem to be a little jumbled, some of them, such as controller, server , are not appropriate. Can you simplify them to highlight the key points?

5. What fields is this paper focusing and closely related of, although with the Journal Not Specified  submission.

6. In Section 3.1 C, what’s the difference between “telecommunication infrastructures” and “telecommunication network element”, are they the same? Would it be better to change the way of expression?

7. The part of the selected tools could be appropriately reduced, it would be better to tell the reasons why they can meet the above requirements in a shorter way.

8. In Section Setup Description, a developed SDN application is mentioned. But there is hardly any introduction about it , at least the framework it is based on, or the basic function implementation process.

9. How to simulate the distance between different nodes, the statement in the article seems unclear, by the functions of Mininet-WiFi or codes you write?

10. In figure 4 and text in Section 6.1, there’s almost no difference between railway and highway. Some specific settings should be adjusted based on the actual situation, otherwise, it is difficult to reflect the innovation and contribution of this research.

11. It would make better experience to modify “Scapy is used in this practical work to create a data packet to indicate critical data communication and to create special messages associated with emergencies or to convey any kind of information to the rail service server or car service server.” in Section 7-2, since it’s too long to read.

12. In Section 7-3, “To demonstrate the video or critical video communication in railways and road co-existence scenarios, a VLC player is used to transmit the video from the train host to the rail server”, does video streaming communication only exist between trains and stations, how about cars,or there’s no difference in this study?

13. In Section 8, “This implies that any traffic between cars to trains and to their assigned service server is disabled”, is communication between trains only in this mode?

14. In Section 8, the part of “If both source and destination IP pairs are in the same network slice, the application checks whether the data packet is tagged with a VLAN ID or not.“, what if the data packet is not tagged with VLAN and source and destination hosts are not connected to the same access point/switch? Are both must connected to the same switch when the data packet is not tagged with VLAN ?

15. In Section 9, it seems that the handover and moving functionality of stations is the same with it of trains/cars, there’s no text or diagram that indicates that they are different. And how to understand the the handover and moving functionality of stations, what’s the basis of this in reality?

16. In Section 9.2, “.Using this method user can start and stop the mobility of the stations but it does not have a pause and start again moving functionality of the stations.”, How to pause? To start it again after stopping for a period of time manually?Or writing codes to restart on time with the help of changing coordinate change?

17. In Section 9.5, are there more test data in other topology and node scale scenarios by iperf?

18. In Section 9.7, what’s the difference between video communicatio and critical video communication, has it been reflected in this paper?

Author Response

(The authors gave the same response as above.)

Reviewer 4 Report

Thanks for submitting your work to MDPI Network!

This paper describes the importance and novelty of emulating the SDN-based co-existence of railways and roads’ telecommunication infrastructure. Then, this paper provides very detailed information on the emulating scenarios and tools, as well as the instructions on setting up and conducting the emulation.

The topic this paper studies has strong potential on improving the railway traffic management systems. This tutorial could become a root of many future research projects on vehicular communication networks. Overall, the paper is well-structured. There are a few places in the paper which can be improved or clarified. Please see details below.

Section 3.1:

Line 102 - 121: It may be conciser to describe R1, R2, R3, R4 and C1, C2 separately. Then you can say T1 = R1C1, T2 = R1C2, etc.

Line 148 - 149: Could you list a few services and applications as examples?

Section 6.1:

Line 333: What’s the unit of “coverage range”? Meters?

Line 350: What’s the unit of “bw”? Mbps?

Section 7:

Table 2: Command 3: it would be better to include “-b <bitrate>” to match your explanation.

Table 4: “To Calculate the Loss in ms”: the unit of loss should be percentage or the number of packets or bytes.

Section 9.1:

Figure 11: It would be nicer and cleaner if your software catches the error and logs it to either a log file or the screen. The stack traces are not informative in this scenario.

Section 9.3:

Table 5: The column headers and row headers are sources and destinations respectively, is that correct? It would be better to clarify it. What’s the meaning of the table’s contents? Instead of Car1, Car2, etc., should they be a simple check mark (as opposed to a cross mark)?

Section 9.5:

According to Figure 26, the link capacity tests of Train1 to RailServer and Car1 to CarServer were conducted one by one. A more meaningful capacity test in the co-existence scenario would be running the two iperf tests simultaneously. That test will show e.g. Wifi’s airtime congestion if they share the same Wifi channel (your setup looks like this case according to Figure 13 and 14).

Section 11:

Line 849 - 854: “After executing the Mininet-WiFi … … to only SUMO integration.” These sentences are discussions of SUMO integration, which would be better to be put in the section describing SUMO, rather than in Conclusion.

Typos and grammars: to name a few:

Line 454: “with a CLI its API is” --> “with a CLI. Its API is”

Line 527: “the controller installed the forwarding rule” --> “the controller installs the forwarding rule”

Line 576: “PAI” --> “API”

Author Response

(The authors gave the same response as above.)

Round 2

Reviewer 3 Report

Thanks for the detailed response to the comments, which generally answers my doubts and gives a better understanding of your work. Up to now, most of the problems have been solved, but there’re still some questions about the following aspects, mainly about the innovation and practical significance of the work, as well as some other details:

1. Could you describe the difficulty of implementing co-existence scenarios with Mininet-WiFi and SDN based slicing, the reason why previous works did not choose to use these tools and make the telecommunication infrastructure shared by the railways and roads?

2. It seems that the railways and highways in this system are generated in the almost same way and consist of the same components except some individual properties such as the different speeds and densities of the nodes that you mentioned. We could hardly tell the significance of distinguishing railways and highways and why it’s so hard to implement co-existence scenarios that no previous work could do it from figures given in this paper. Maybe the difference of their severs is the key point, please provide some additional explanations.

3. What do you mean by “station”, is it different from the train/car? How to understand the movement of the station, a moving station?

I hope that you could make some explanations and supplements on above issues.

Author Response

Dear Sir/Mam,

Thanks a lot for your feedback and please see the attachment for the asked questions.

Once again thank you.

BR,

Radheshyam Singh
